# An Overview of Exertional Heat Illness in Thoroughbred Racehorses: Pathophysiology, Diagnosis, and Treatment Rationale

**DOI:** 10.3390/ani13040610

**Published:** 2023-02-09

**Authors:** Meg Brownlow, James Xavier Mizzi

**Affiliations:** 1Racing Australia, Druitt Street Sydney, New South Wales, Sydney 2000, Australia; 2Department of Regulation, Welfare and Biosecurity Policy, The Hong Kong Jockey Club, Sha Tin Racecourse, Sha Tin, Hong Kong

**Keywords:** exertional heat illness, thoroughbred racehorses, case definition EHI, treatment options EHI, pathophysiology EHI

## Abstract

**Simple Summary:**

Exertional Heat Illness (EHI) and its associated fatal form of heat injury, heat stroke (HS), is not a new disease. It has been described as the oldest known medical condition and its occurrence in humans dates back more than 2000 years. The condition occurs when an individual is unable to adequately dissipate the metabolic heat produced by physical exertion, and the resulting elevation of core body temperature can cause damage to multiple organs. EHI in human subjects is a particular threat to the health and safety of athletes, both elite and recreational, military personnel, firefighters, and outdoor labourers. Similarly, EHI/HS in horses is not new, but progress in managing the condition has languished due to the relative lack of information provided in the veterinary literature. There is, however, an important similarity between racehorses and human athletes, in that both dissipate heat chiefly by evaporative sweating. Therefore, much that has been published on EHI in the human scientific literature can also serve to enlighten us about the condition in horses. To ensure the welfare of racehorses, the authors have described EHI in detail. It is imperative that veterinarians working at the racetrack can recognize the earliest clinical signs, understand its mechanism of action and the rationale for practical treatment options

**Abstract:**

Exertional heat illness (EHI) is a complex medical disease. The thoroughbred (TB) racehorse is at considerable risk because of the intensity of its exercise activity and its high rate of metabolic heat production. The pathophysiology of EHI can combine aspects of both the heat toxicity pathway and the heat sepsis or endotoxemic pathway. Treatment regimes depend upon the detection of earliest clinical signs, rapid assessment, aggressive cooling and judicious use of ancillary medications. Ice-cold water provides the most rapid cooling, consistent with the need to lower core body temperature before tissue damage occurs. Research into EHI/HS by inducing the condition experimentally is ethically unjustifiable. Consequently, leading researchers in the human field have conceded that “most of our knowledge has been gained from anecdotal incidents, gathered from military personnel and athletes who have collapsed during or following physical activity, and that retrospective and case studies have provided important evidence regarding recognition and treatment of EHI”. The authors’ review into EHI shares that perspective, and the recommendations made herein are based on observations of heat-affected racehorses at the racetrack and their response, or lack of response, to treatment. From 2014 to 2018, 73 race meetings were attended, and of the 4809 individual starters, signs of EHI were recorded in 457. That observational study formed the basis for a series of articles which have been published under the title, ‘EHI in Thoroughbred racehorses in eastern Australia’, and forms the background for this review.

## 1. Introduction

Exertional Heat Illness (EHI) occurs in all species where strenuous physical exertion takes place. It has been documented in human athletes, military personnel, firefighters and outdoor labourers [1], in racehorses and sporting horses [2,3,4], racing camels [5], and in working, sporting and even recreational dogs [6,7,8]. The major symptoms of hyperthermia manifests as central nervous system (CNS) dysfunction which is typical across species, and EHI represents a significant welfare issue because it can cause significant morbidity and even fatality if allowed to progress untreated [9]. Despite efforts in the human athletics field to provide prevention and educational initiatives, recent reviews of the literature reveal little or no change in the annual number of deaths associated with EHI [10]. Scholars in the human EHI field maintain that prevention of EHI is difficult because of its idiosyncratic nature and considerable individual variability in those affected. However, it can be treated with positive outcomes, but only if recognized early [1,11]. Historically, one of the major issues in athletics and the military has been lack of knowledge of the condition, resulting in affected individuals being ignored or not treated satisfactorily within the critical timeframe to minimize hyperthermia-related tissue damage [12]. 

The aim of this article is to clarify exertional heat illness in the TB racehorse for those veterinarians working at the racetrack, help them to recognize EHI in its earliest stages, understand the various pathophysiologies involved, and achieve best practice outcomes in the treatment of heat-affected horses. The following section deals with the way racehorses cope with the challenge of performing strenuous exercise in hot and humid conditions, essential knowledge in relation to the pathogenesis of EHI. 

## 2. The Thoroughbred Racehorse: Superb Athleticism and Thermoregulatory Specialisation

### 2.1. Superb Athleticism Results in a Hyperthermic Response to Exercise

During a race, the principal source of heat acting on a horse’s body is the metabolism associated with muscular contraction, which is largely independent of the thermal environment. Production of metabolic heat increases abruptly at the onset of exercise and is commensurate with the intensity and duration of the specific exercise activity [13]. Much of the heat generated during this stage is stored and distributed throughout the body by blood flow through the vascular system and by tissue conduction, resulting in elevation of core body temperature [14]. The seminal work of Hodgson and colleagues [15] showed that the racehorse had the highest rate of heat production of all sporting horses at 1250 kJ/min, and they theorised that if all that metabolic heat was stored, the core body temperature of the TB racehorse could rise by 0.8 °C per min and a temperature of 42.0 °C could easily be reached. Such temperatures have frequently been reported in experimental treadmill studies without clinical manifestations of EHI [16,17]. 

For a long time, a core body temperature >40.5 °C was relied upon in the human field to make a definitive diagnosis of EHI or heat stroke (HS) [18]. Recently, however, the importance of a temperature threshold has been questioned by the American College of Sports Medicine in favour of the recognition of CNS dysfunction. This was because athletes performing high intensity physical activity in both warm and hot weather were commonly found to have core temperatures above 40.0 °C with no apparent ill effects [1]. The critical temperature for EHI in horses is unknown, although various treadmill studies have cited core temperatures between 42.0 to 43.0 °C without any obvious clinical symptoms of EHI [19,20]. Anecdotal evidence [21] suggested that 43.5 °C was commonly observed in horses with overt clinical manifestations of EHI such as ataxia, and temperatures of 44.0 °C observed with collapse. Shapiro [22] experimentally induced HS in dogs and found that the critical core temperature for heat injury was 43.0 °C. Animals whose temperature stayed below this level showed no adverse clinical or clinicopathological effects, but dogs whose temperatures reached 43.0 to 44.0 °C had substantial clinical manifestations of heat illness and mortality rates ranging from 50% to 66%. At temperatures above 44.0 °C the mortality rate was 100%.

These findings suggest that we may need to rethink temperature levels for thermal injury in TB racehorses. Most treadmill studies have used 42 °C as a critical point, but this has not resulted in any reported cases of EHI [13,14,15,16,17,19,20]. In fact, to quote Dr. David Poole, a researcher with extensive experience in treadmill research with horses [23] (p. 7), “body core temperature can rise extremely quickly in the horse and routinely achieves values of more than 42.0 °C without any discernible adverse effects”. The authors are of the view that such a temperature is probably normal for TBs performing strenuous exercise, and the traditional ‘line in the sand’ benchmark temperature extrapolated from human studies, between 40° and 42 °C, is probably not appropriate. It may, in fact, provide misleading results, especially for cooling studies, because the horses are not heat affected and have normal cardiovascular physiology.

### 2.2. Thermoregulatory Specialisation: Evaporative Heat Loss by Sweating and Respiratory Evaporative Heat Loss (REHL) from the Powerful Thermo-Effectors of the Upper Respiratory Tract

*Heat losses, convective and evaporative*: Heat is lost to the environment by the combination of the physical heat exchange processes of convection, radiation and evaporation. Both convective and evaporative heat exchanges depend on the movement of air over the skin surface and the respective temperature and humidity gradients between skin surface and environment. Convective heat exchange will result in heat loss if air temperature is lower than skin temperature, and heat gain if air temperature is higher than skin temperature. During racing these effects can be magnified by air flow over the body surface resulting from the combination of body movement through the self-facing air mass [13,24] and environmental wind. 

Heat exchange by convection and evaporation are closely linked so that if there is minimal convective heat exchange or where there is actual heat gain, temperature regulation becomes reliant entirely on the evaporation of sweat. In horses this takes place through vaporisation from the skin surface and from the mucosal surfaces of the respiratory tract [13].

*Evaporative heat loss* via *sweating.* It is well documented that the horse is a robust sweater and probably demonstrates the highest sweat rates in the animal kingdom [13], with sweat glands all over its body in both the haired and relatively hairless skin [25]. Horses also possess an unusual protein-rich sweat, a major component of which is latherin, thought to act as a surfactant or wetting agent to facilitate the translocation of sweat water from the skin to the surface of the hair for efficient evaporative cooling [26]. Some authors have used the term ‘thermal window’ [27] to describe certain areas of the body which are ‘hot spots’ for heat exchange. For instance, the ears of elephants dissipate heat most effectively due to the huge surface area and high concentration of blood flow. Figure 1 shows an albino horse immediately after racing in warm/hot conditions. Note that almost the entire horse is pink, suggesting increased blood flow. It is suggested that most of the racehorse’s skin surface may function as a thermal window for heat transfer. Figure 2a,b shows the skin surface in two different horses some 10 to 15 min after racing and demonstrates the incredible redistribution of blood flow to the skin for heat exchange. 

### 2.3. Respiratory Evaporative Heat Loss (REHL) from Powerful Thermo-Effectors of the Upper Respiratory Tract

The upper respiratory tract in the horse functions as a powerful and efficient evaporative heat exchanger. It consists of a series of turbinate bones covered by a blood-rich mucosal surface which contains numerous arterio-venous anastomoses (AVAs). It has been estimated that as much as 30% of the metabolic heat produced during strenuous exercise can be dissipated by this pathway [28] (see Figure 3). 

### 2.4. In the Post-Exercise Period and in Mild Thermal Conditions Normal Ventilatory Patterns Result in Evaporative Heat Loss

This is achieved by inhalation of air over the moist mucosal surfaces of the upper respiratory tract, so that the air is humidified, and the mucosal surfaces are cooled. The air that passes down into the lungs becomes saturated with water vapour at body temperature and as it is exhaled it passes back over the cooler mucosal surfaces of the nose where it loses a percentage of its water content, essentially returning some of the water and the heat collected during inhalation. 


**
*In Warm to Hot Environments, Hyperthermic Responses Accelerate the REHL Mechanisms*
**


After exercise in these conditions there is increased mucosal blood flow, assisted by the AVAs, and this warms the mucosal surfaces more rapidly following inhalation. This results in less water and heat being retrieved during exhalation and allows more heat to be dissipated to the environment. 


**
*Panting Is the Next Stage*
**


The respiratory rate is substantially increased but air flow is essentially in and out of the upper respiratory tract, so that evaporative heat loss is maximized without deleterious changes to acid-base balance [28]. Two studies in horses support the argument that horses can pant effectively [16,20]. The rate of evaporative heat loss is, however, dependent upon the water vapour pressure of the air and the maximum ventilation rate. If water vapour pressure of the environment increases, the ability to evaporate heat via both the respiratory route and the skin surface will be impaired [13]. At the racetrack it is easy to recognise when a horse is panting. The respiratory rate increases to over 120 breaths per min, the nostrils are widely dilated (see Figure 4), but despite the high ventilation rate there are minimal thoracic excursions and horses appear to adopt a rocking motion because of their rapid breathing.


**
*Additional Heat Loss Effectors: Venous Transporters from the Face and Scalp*
**


The REHL results in cool venous blood draining from the scalp and face via the angularis occuli (orbital) vein and the facial vein respectively (see Figure 3 and Figure 4). The former drains into the cavernous sinus, whilst the latter drains into the jugular vein. The facial-jugular pathway has been shown [29] to contribute to general body cooling by discharging blood up to 3.0 °C cooler than the core body temperature of 42.0 °C directly into the right heart. Given that there is an extremely high rate of blood flow into the jugular vein this pathway could deposit a substantial amount of cooler blood into the central circulation each min resulting in a significant whole-body cooling effect and representing a most significant and largely underestimated contribution to cooling in horses.

### 2.5. The Consequences of Strenuous Exercise in the Heat: Changes to the Distribution of Cardiac Output at Rest, during Maximal Exercise, and during Recovery

The cardiovascular system can only supply a finite volume of blood to the organs of the body and competing demands for supply result in a redistribution of blood flow. This is shown diagrammatically in Figure 5. During exercise, cardiac output favours contracting skeletal muscle and the heart, while splanchnic and renal portions ‘lend’ their share of blood flow to areas of greater need. This has important ramifications for the gastrointestinal tract (GIT) because, depending upon the intensity and duration of the exercise activity, the GIT will be subjected to hypoperfusion, causing increased permeability of the mucosal barrier and the possible release of endotoxins into the general circulation [30].

### 2.6. Practical Implications of Thermoregulation in the Recovery Period: Conflict between Thermoregulation and Maintenance of Circulation, and its Relationship to the Pathogenesis of EHI

One of the most prominent scholars of thermoregulation [13] has compared the recovery period after strenuous exercise in the heat to a state of acute haemorrhage. Physiologically, cardiac output has been redistributed to the skin for heat dissipation and because the venous side of the circulation is distensible and highly compliant, a larger volume of blood is sequestered there, causing a reduction in effective blood volume [13]. This is referred to as ‘redistributive hypovolemia’. Figure 6 shows the medial side of a hindleg in a horse 15 min after racing. Because veins are more compliant than arteries they dilate when there is increased blood flow, resulting in a significant increase in the volume of blood retained in the peripheral circulation.

The superb thermoregulatory capacity of the TB racehorse will enable it to compensate for such changes normally, but if the individual has emerging EHI, such cardiovascular alterations will have a significant effect on the capacity to cool. This is because the rate of convective heat transfer from the body core to the skin surface is proportional to the product of skin blood flow and the temperature gradient between core and skin [13]. If overall cardiac output is diminished, blood flow from the core to the skin will be reduced, placing greater reliance on the temperature gradient to assist cooling. Individuals so affected present clinically with ‘slow recovery’ times. They are panting, respiratory rates are elevated (>120 per min), with nostrils widely dilated. Capillary refill times are slow and there is evidence of dehydration, based on skin turgor elasticity tests. Under different circumstances these clinical findings might necessitate an intravenous fluid infusion; however, these effects are transitory. Nevertheless, effective cooling is mandatory and cold water will be most effective for promoting internal heat transfer. Cardiovascular stability will usually return within 30 to 40 min after racing [32]. For the track veterinarian, a knowledge of the physiology of the recovery period is of crucial importance because this is the time when horses are at greatest risk of EHI, and it is sometimes difficult to discern between physiological changes resulting from an arduous run and those due to emergent EHI.

## 3. The Pathophysiology of Exertional Heat Illness in Thoroughbred Racehorses

Strenuous exercise, especially during heat stress conditions, causes increases to metabolic heat production and a hyperthermic response, although the critical temperature at which EHI occurs tends to be inconsistent. In human subjects, experts are now recommending the presence of CNS dysfunction as the main diagnostic criterion to identify EHI [33] (p.1340). There are two suggested EHI/HS pathophysiological pathways, one due to hyperthermia and the other due to the initiation of inflammation,,various combinations of which may contribute to the clinical condition, depending upon the additional influences of environmental (extrinsic) risk factors and host (intrinsic) risk factors. 

### 3.1. The Heat Toxicity Pathway (see Figure 7)

The direct effect of heat on cells is characterised by the denaturation of protein, causing irreversible damage. All cell components are affected, including the cytoskeleton, membranes, and nucleus, leading to decreases in cell viability and cell death. Thermal injury in general results in damage to vascular endothelium, initiating coagulopathies which can progress to cause widespread intravascular coagulation. A characteristic pathological finding is microthrombi deposition in the liver, lungs, heart, and kidney, creating widespread organ damage and ultimately multi-organ failure (MOF) [34].

**Figure 7 animals-13-00610-f007:**
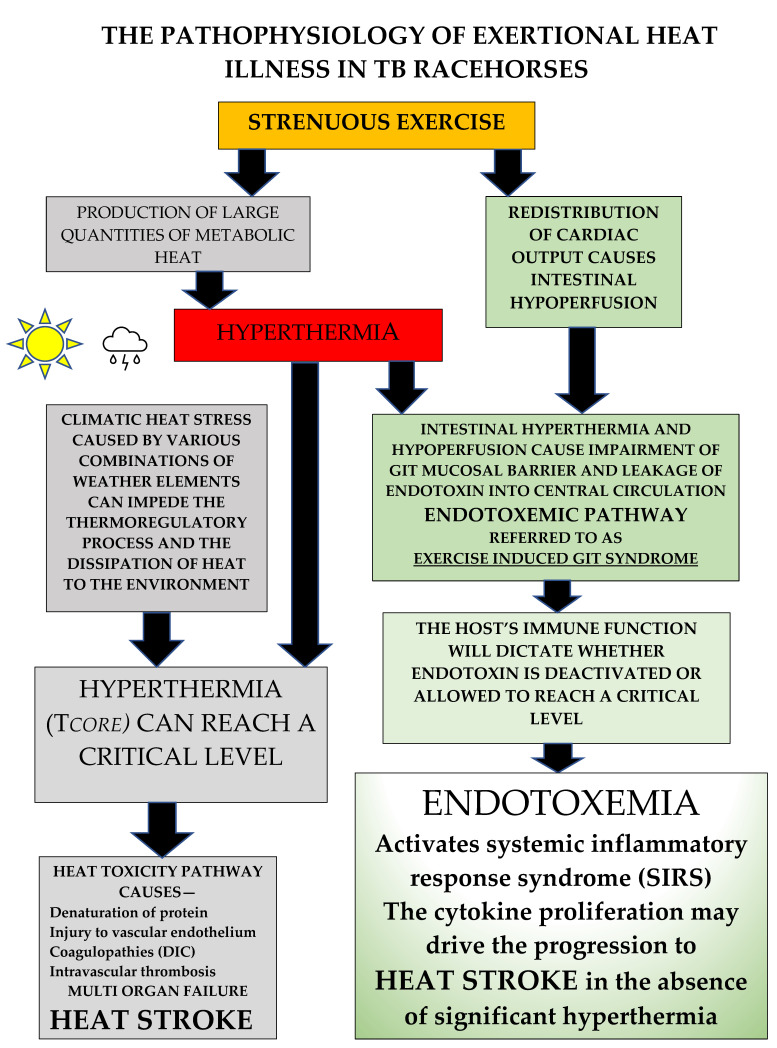
There are two proposed pathophysiological pathways for EHI. The strenuous exercise of racing produces a large amount of metabolic heat, most of which is stored, resulting in hyperthermia. If an individual’s ability to thermoregulate is impaired by high levels of heat and humidity, hyperthermia may reach critical levels. This can trigger the heat toxicity pathway, potentially leading to heat stroke with multi-organ failure. Strenuous exercise also causes a redistribution of cardiac output. Heat and intestinal hypoperfusion combine to cause impairment of the intestinal mucosal barrier, allowing leakage of endotoxin into the central circulation. Limitation of this endotoxemic or ‘heat sepsis’ pathway is dependent on the host’s immune system, which if functioning normally can deactivate endotoxin. If this does not occur, endotoxin can reach a critical level and may activate the systemic inflammatory response syndrome (SIRS), stimulating the production of cytokines, which might actually drive the condition and are thought to mediate many of the adverse consequences of the EHI/HS syndrome.

As previously stated, the degree of heat-related tissue damage is dependent upon temperature and time. If the time to reach a certain level of thermal injury is plotted as a function of temperature, a biphasic linear response is seen that has common slopes across all species and mortality endpoints. Thus, for most mammalian tissues the body temperature at which the curve changes slope is between 43.0 and 43.5 °C. According to Goldstein [35], 240 min at 42 °C causes the same level of thermal damage as 60 min. at 43 °C, 30 min. at 44 °C, or 15 min. at 45 °C. This has important ramifications for the management of affected individuals. Most racehorses compete, are cooled, and then left in their tie-up stalls for a period before being transported to their point of origin, which in Australia can sometimes be several hours away from the racetrack, and horse transporters are not always air conditioned. If a particular horse has not cooled adequately, it is quite possible that a temperature of 41.0 to 42.0 °C may persist for some hours and theoretically result in significant thermal damage to tissue at temperature levels not usually considered to be pathologically significant. 

The traditional viewpoint is that all cases of EHI occur when horses or human athletes perform strenuous exercise in conditions of extreme heat/humidity. Clearly, a single overwhelming exposure is responsible for many cases, but a recent epidemiological study in TB racehorses in eastern Australia [36] revealed that environmental conditions were responsible for only 43% of the EHI cases recorded. This has significant implications for understanding EHI, suggesting that factors other than environmental ones may be involved, which might have compromised the individual’s ability to regulate heat strain and control the exercise-associated hyperthermic response.

### 3.2. The Endotoxemic or ‘Heat Sepsis’ Pathway (see Figure 7)

By itself, the heat toxicity pathway could not fully explain why in some instances the pathology associated with fatal heat stroke cases was more characteristic of sepsis, like that seen in acute bacterial infections. In 2002, clinical researchers Bouchama and Knochel [37] proposed that EHI/HS could be represented by another pathophysiology, which they termed the ‘heat sepsis’ or endotoxemic pathway. This was supported by others who experimentally showed in human subjects that strenuous exercise, especially under adverse environmental conditions, caused intestinal hypoperfusion and hypoxemia, disrupting the integrity of the mucosal gastrointestinal (GIT) barrier [38] and allowing endotoxins to gain access to the central circulation. It was suggested that under certain circumstances and particularly in cases of immune dysfunction, endotoxemia could reach critical levels and trigger a systemic inflammatory response syndrome (SIRS) characterized by cytokine proliferation, where the latter became the major driver for the progression of EHI. Importantly, within this description there was a lack of inclusion of a specific T*c* value above which this occurred, reflecting the reports of substantial variability for temperature measurement in clinical cases [39].

In the last few decades this model of disease has become much more relevant, as has the importance of the entire GIT microbiome in maintaining mucosal integrity and overall gut health. Many researchers view the endotoxemic pathway as a probable major contributor to the pathophysiology of the EHI/HS disease complex in humans, referring to it as ‘the exercise-induced gastrointestinal syndrome’ (EIGS) [40,41]. When reviewing the literature in this area it becomes apparent that there are similarities between humans and horses in their physiological response to EHI and its clinical manifestations. In ponies, McConaghy [42] showed that there was an exercise-related decrease in intestinal blood flow, which was exacerbated by both heat and exercise intensity. Secondly, Baker [43] showed that when TBs raced there was an associated elevation in endotoxin levels and, as observed in human athletes, anti-endotoxin antibody was also present, functioning as a potential brake on its accumulation and thereby enabling heat tolerance. Finally, in another study, TB racehorses were exercised to absolute fatigue [44]. Endotoxin levels increased threefold at the point of fatigue and gradually decreased over the next two hours. Most importantly, however, it was noted that the exercise bout initiated a pro-inflammatory response with cytokine proliferation typical of that seen in human subjects with exertional heatstroke [37]. These results are indicators that a similar response to strenuous exercise may exist for TB racehorses, but further research is warranted. An important point is the extreme sensitivity of both the equine and human species to the effects of circulating endotoxin, and the similar cytokine proliferation responses in both experimental infusion models and naturally occurring clinical cases of endotoxemia [45].

From extensive field experience, the authors confirm that EHI in TB racehorses is often unpredictable, with frequent occurrences in cooler weather conditions, meaning that potentially, EHI can occur on any day that a racehorse performs strenuous exercise, a situation which does not conform to the heat-centred theory. There are also cases where horses exhibit a slow-to-cool response, creating questions about the underlying pathophysiology. If cooling modalities are optimal the clinical condition should be improving, but recently, Leon and Helwig [39] described HS cases in humans where rapid cooling initiatives and other resuscitative therapies were inadequate to prevent tissue injury. This was attributed to the heat-induced SIRS and its associated cytokine proliferation, rather than the direct effects of hyperthermia. Those authors [39], ascribed many of the adverse consequences of the HS syndrome to the peripheral and CNS actions of cytokines. EHI should therefore be regarded as a more complex medical disorder, in which the involvement of thermoregulatory and inflammatory pathways is sometimes mixed, and while hyperthermia may be the trigger, it may not always be the ultimate driver of the condition [33,39,40].

### 3.3. Central Nervous System Dysfunction Is a Common Feature in All EHI Clinical Cases and Should Be Considered the Main Diagnostic Criterion [33]

Sharma and Hoopes [46] have clarified the CNS pathophysiology of heat stroke in extensive studies using rodent and dog heat stroke models, and this provides the basis for understanding the earliest clinical signs and their corresponding CNS pathology, which is similar in most mammals (see Figure 8). Firstly, there is a hyperthermia-induced reduction in cerebral blood flow, leading to cerebral ischemia. In human subjects this is responsible for the earliest clinical signs, which are usually very vague and can include confusion, irritability, extreme restlessness, combativeness, and substantial headache. If the hyperthermia progresses, serotonin initiates an increase in permeability of the blood-brain barrier (BBB) which allows leakage of plasma proteins from cerebral capillaries, causing cerebral oedema [47].

Escalating CNS dysfunction ensues, and if treatment is delayed, may progress to an associated neuronal injury, specifically involving the cerebellum. This is followed by ‘heat stroke’ and involves profound clinical signs of CNS dysfunction. There is delirium and stupor, with individuals being unaware of their surroundings. End-stage HS is characterized by an oedematous, swollen, brain in a closed cranial compartment. This causes compression and cellular damage to vital structures, eventually resulting in collapse, loss of consciousness, coma, and ultimately death. Numerous researchers consider that elevated levels of CNS serotonin are the key driver in the hyperthermia-associated increase in permeability of the BBB, which leads directly to progressive cerebral oedema and neuronal injury, and possible coma and death [34,48].

### 3.4. Case Definition of EHI in TB Racehorses

Recently, a consensus of expert opinion in the human EHI/HS field has stated that “the associated levels of CNS dysfunction are the true and most reliable indicators of EHI in the presence of any given level of hyperthermia”, which in human subjects can be as low as 38.5 and as high as 47.0 °C [1,33]. The case definition for EHI in racehorses is based on descriptors for CNS dysfunction on a continuum from Level 1 to 4 [32]. Although individual animals can enter at any level, they may also escalate between levels if treatment strategies are not effective (see Figure 8). EHI commonly occurs in the immediate post-race period and horses usually display substantial levels of distress with elevated heart and respiratory rates. They may have adopted a panting type of respiration, most are sweating profusely, and in some cases sweat is dripping off their bodies. Note that rectal temperatures were not taken at the racetrack by the authors due to the extreme risk posed by ataxic and disoriented horses.

***Level 1: case definition descriptors: (see***Figure 8 and Figure 9***).***


Level 1 horses display the earliest recognizable signs of CNS dysfunction, with extreme restlessness and agitation, and an irritability which is not typical for that individual. There is often head nodding or head shaking; the horse cannot or will not stand still and is difficult to restrain. Level 1 symptoms can resolve rapidly following routine cooling interventions but also may escalate if the cooling process is inadequate or if other processes are operating. It is important, therefore, to identify these horses and prioritise their cooling, because they are at increased risk of EHI. As a general principle, it is important to encourage horse handlers to use the cooling devices, firstly to acquaint them with their operation, then to educate and inform them concerning EHI, and encourage them to take personal responsibility for horse welfare. 

***Level 2: case definition descriptors: (see***Figure 8 and Figure 10***).***


In Level 2 horses, neurological dysfunction is more obvious. There is often a ‘kicking out’ type of behaviour. This is not in response to any particular stimulus. It can be violent and continuous or spasmodic with associated levels of irritability and agitation, tending to escalate. This is often misdiagnosed by handlers and veterinarians new to the track as a colic episode. At Level 2, behaviours can be extremely unpredictable. The author (MB) has witnessed the simple placement of a towel across the back of a horse causing it to flip over backwards and fracture its wither and poll.

***Level 3: case definition descriptors: (see***Figure 8 and Figure 11***).***

These horses can display many and varied bizarre neurological behaviours, and misdiagnosis at this stage is still common. Horses have ‘altered mentation’ and this has been anecdotally described as appearing to be ‘spaced-out’, ‘glassy-eyed’, or having a ‘vacant expression’. They are extremely disorientated, may have a head tilt, may lean to one side, and are commonly described as being ‘wobbly’ by their handlers. At Level 3 there are always varying levels of ataxia, horses can walk forward, then stop, rear, and throw themselves over backward. They can run into obstacles, into people and into fences and at this stage are an extreme risk to themselves and to their handlers. A strange hind limb lameness might also appear, referred to by the authors as ‘the broken-leg syndrome’. In these cases, horses are completely non-weight bearing on the affected leg, and hop along using their good leg, but the condition completely resolves with cooling and other ancillary treatments. This is, however, another major cause of veterinary misdiagnosis. 

***Level 4: case definition descriptors: (see***Figure 8 and Figure 12***a–c).***


These horses present with substantial levels of CNS dysfunction. In almost all cases, horses are extremely ataxic, tend to be unaware of their surroundings and are very disorientated. They fall down repeatedly, then stagger to their feet and fall again. They can collide with objects and people and may demonstrate plunging behaviours. Such horses are at extreme risk of injuring themselves and the people around them. Injuries are common, with fractures to the poll, withers, and head. In extreme circumstances, collapse may follow with loss of consciousness and possible death.

## 4. Rationale behind Treatment Strategies for EHI in TB Racehorses

### 4.1. Drugs That Target CNS Dysfunction

Detomidine hydrochloride **^a^**—a synthetic alpha-2 adrenoreceptor agonist with dose-related sedative and analgesic effects in non-heat-affected horses. This family of drugs is known to display serotonin antagonist activity, serotonin being the neurotransmitter implicated in the cerebral pathophysiology of heat stroke [46,47,48,49].

***Dose rate:*** 0.5 mL to 1.0 mL per 450 kg as required to abolish CNS symptoms.

**a** Dormosedan—(10 mg/mL) Zoetis, Parsippan, New Orleans United States.

Originally out of sheer necessity, the first author (MB) used detomidine to restrain racehorses from inflicting harm on themselves and endangering their handlers. Surprisingly, after administration to EHI-affected horses there was a complete reversal of all CNS signs within 2 to 3 min and no signs of the typical detomidine-associated sedative response. Since that time, detomidine has been used routinely for EHI cases. It has provided consistent and reliable outcomes and has become the cornerstone of treatment.

***Proposed mechanisms of action:*** In the seminal body of work by Sharma [46,47,48], using rat and dog experimental heat stroke models, hyperthermia was shown to be instrumental in initiating blood–brain–barrier dysfunction, leading to a vasogenic cerebral oedema, and the neurochemical transmitters dopamine and serotonin were associated with its pathogenesis. To support the rationale for using detomidine, an in-depth pharmacological study of medetomidine (a closely related methylated derivative) is cited [49]. In the study, this family of drugs was found to depress the turnover of biogenic amines such as noradrenaline and dopamine in the brain, but most importantly, there were significant decreases to serotonin levels. The neurotransmitter serotonin has been implicated as a key driver of the cerebral pathophysiology associated with the EHI/HS condition [48,49]. Another researcher [50], commented that drugs which act as serotoninergic nerve depletors or receptor antagonists were able to protect against cerebral heat stroke reactions. This may explain the excellent results in heat-affected horses observed after using a single dose of detomidine. It must be emphasized, that horses administered detomidine for EHI show no apparent sedation but revert to calm, normal behaviours and there have been no deleterious consequences associated with its use [32].

### 4.2. Drugs That Target Endotoxemia 

Flunixin meglumine **^b^**—a non-steroidal anti-inflammatory drug (NSAID). Flunixin blocks the action of cyclooxygenase (COX) enzymes, which convert arachidonic acid to prostaglandins (PGs) and thromboxane. PGs are metabolic end products, synthesized in response to a variety of external stimuli, including thermal injury, and regulate many physiological responses, including cytokine proliferation [39]. Flunixin is considered the NSAID of choice for the treatment of equine endotoxemia [51].

***Dose rate:*** 1.1 mg/kg as a single intravenous injection.

**b** Flunixon (50 mg/mL)—Norbrook Inc., Overland Park, Kansas, United States.

One of the suggested pathophysiological pathways for EHI/HS involves endotoxemia or ‘heat sepsis’ [37,38,39,40,41,43,44,45]. Leakage from a heat-affected and exercise-intensity-associated hypo-perfused GIT allows endotoxin to accumulate in the central circulation. Endotoxin is thought to be responsible for the stimulation of reactive metabolites, including cytokines, under heat stress conditions. Endotoxin and cytokines have been implicated as key mediators for the heat-induced SIRS, and correlative data have shown high circulating concentrations of these substances in human and animal HS models [39].

Flunixin meglumine is an NSAID which exhibits anti-inflammatory, analgesic, and antipyretic properties, and represses cytokine proliferation in human and animal models [39,52,53]. The NSAIDs have been extensively studied in horses, and the results of comparative drug studies have clearly indicated flunixin meglumine as the most effective in preventing endotoxin-induced synthesis of PGs and thromboxane, and ameliorating the clinical signs associated with endotoxemia in affected horses [52].

The only potential problem with the use of NSAIDs lies in their capacity to cause nephrotoxicity and GIT side-effects, such as delayed intestinal mucosal repair [53]. Horses found to be most at risk of renal damage, however, are those given excessive doses, and those already experiencing significant levels of dehydration at the time of administration. Dehydration is relatively uncommon in our racing jurisdiction, where most horses compete in sprint events of short duration. Whilst a redistributive hypovolemia does occur post-race, it normalises very quickly with proper cooling strategies, and the risk entailed with a single dose of flunixin, if weighed against the adverse outcomes of EHI, is considered justifiable. Because of the possibility of a contribution from the endotoxemic or inflammatory pathway to a heat-affected individual, the authors strongly recommend administration of both detomidine and flunixin in quick succession. This combination has been found to provide positive and reliable outcomes.

### 4.3. Drugs Shown to Prevent the Synthesis, Release, or Action of Specific Reactive Metabolites in the Endotoxemic Pathway

The corticosteroid dexamethasone sodium phosphate **^c^** is a glucocorticoid with potent anti-inflammatory and immunosuppressive activity and is used as a therapeutic agent in horses for a variety of clinical conditions.

***Dose rates****:* from 0.2 to 0.5 mg/kg for routine anti-inflammatory indications, but up to 2.0 mg/kg for septic shock (endotoxemia) as a single bolus intravenous injection. High dose rates may potentiate the onset of laminitis in adult horses [51,52,53], and with this concern in mind, dexamethasone needs to be used with caution at the higher dose rates. Recommended dose rates for heat-related neuroprotection in dog and cat heat stroke models [46] are in the vicinity of 4 mg/kg, and best results were achieved when the drug has been administered prior to the heat injury. 

**c** Dexapent—5 mg/mL Troy Laboratories, Glendenning, New South Wales, Australia.

Historically, dexamethasone has been the drug of choice for many equine practitioners presented with EHI cases at the racetrack, and anecdotal reference to its administration appears frequently [2]. Dexamethasone is thought to specifically inhibit the activity of phospholipase A**_2_**, reducing the liberation of arachidonic acid and blocking the downstream macrophage synthesis of cytokines [54,55]. However, observations specifically in dog and rat heat stroke models show that these and other beneficial effects tend to be at the higher dose rates, and only occur if the drug is administered before the onset of endotoxemia, and in the case of HS, before the onset of hyperthermia. [46,47,48]. 

Recently, dexamethasone efficacy was tested in a model of heat stroke using baboons, but despite being administered prior to the heat stress and during cooling, protection against the lethal effects of heat injury were not realised [56]. Similarly, a well-controlled clinical trial in ponies showed that the effects of endotoxin were less severe and survival times longer when ponies were treated with flunixin, in comparison to dexamethasone or prednisolone [57]. The case for the use of dexamethasone in heat-affected horses is not convincing [46,47,48,52,53,56].

The first author (MB) has found the use of dexamethasone at the racetrack extremely disappointing because horses did not seem to improve after its administration. Interestingly, researchers [57] from one study in ponies commented that the steroids (dexamethasone at 2 mg/kg) took substantially longer to reduce prostaglandin synthesis. The first author (MB) only uses dexamethasone in combination with detomidine and flunixin in critical emergency situations involving Level 3 and 4 horses, but cannot recommend it as a routine, stand-alone medication for EHI [32].

### 4.4. Therapies Which Target the Increase in Core Body Temperature

#### 4.4.1. Cooling Strategies Which Might Target the Brain

A hot brain has adverse consequences. Clinical manifestations of EHI/HS in all species appear as central nervous system disturbances, the severity of which tend to closely parallel the cerebral pathophysiology(see Figure 8) [46,58]. The temperature of the mammalian brain is determined by the rate of heat production by the brain cells, the rate of blood flow through the brain, and the temperature of the blood supplying the brain. Strenuous exercise during heat stress conditions causes activation of the brain, increasing its heat production. The exercise-associated hyperthermic response in racing not only reduces cerebral blood flow, creating cerebral ischemia [46], but also perfuses the brain with hot blood [58]. In the animal species that have been most thoroughly studied, researchers have found that cerebral arterial blood has the greatest potential to produce significant changes to brain temperature [59]. In all mammalian species, paired carotid arteries pass up the neck and divide into internal and external branches. The internal carotid artery then traverses the cavernous sinus at the base of the brain joining with the vertebral arteries to form the circle of Willis from which major arteries arise to perfuse all parts of the brain.

Because the temperature of the brain is primarily defined by the temperature of the incoming arterial blood, cooling the carotid artery might provide a rational avenue for cooling the brain. The first author (MB) has developed a cooling collar as an adjunct to whole-body cooling strategies. The ‘one size fits all’ collar is designed (see Figure 13) to provide a relatively large surface area cooled by crushed ice on either side of the neck. This creates a ‘heat sink’ effect around the carotid artery and jugular vein and in the surrounding tissues. The neck of the horse is relatively long, and the artery and vein have a most superficial position, with closest juxtaposition in the upper third of the neck. Three elastic straps provide adequate pressure around the horse’s neck to hold the collar in place. Although the collar can also accommodate commercially available chemical cold packs, research has shown that ice has a greater cooling capacity and tends to remain colder for longer [60]. The ice collar is usually applied whilst whole body cooling is taking place or afterwards to prevent ‘rebound hyperthermia’. Signs of EHI might not yet be apparent but it is used in such cases as a precautionary treatment strategy, usually for a period of 20 to 30 min, until the horse has shown improvement or all the ice has melted. 

#### The Cooling Collar May Provide Several Benefits

Firstly, it may be cooling the blood in the carotid artery. Cooling collars have been used in human athletes and military personnel, but rather to maintain performance levels during heat stress conditions [61]. Palmer [62] showed that cooling the neck region during high intensity exercise attenuated the rise in brain temperature and improved the individual’s perception of thermal strain. Participants could tolerate high core body temperature and higher exercise-associated heart rate when their necks were cooled. It was documented that the application of ice packs to the lateral surface of the neck could reduce temperatures by between 0.2 °C and 0.5 °C. Secondly, it has been shown experimentally [63] that cooling the artery itself induces dilation of the vessel, which has significant downstream effects, increasing the actual blood flow and perfusion of the brain. Thirdly, the fact that the artery and vein are in close juxtaposition in the upper third of the horse’s neck implies that blood may also be cooled in the jugular vein. If this occurs, cooler blood is entering the right heart for eventual transfer around the body, possibly contributing to whole body cooling. Finally, some areas of the body, such as the neck and hands in humans, have been identified as being particularly sensitive to thermal sensation and able to provoke feelings of perceived thermal comfort/discomfort. This is referred to as allesthesial thermosensitivity, and in heat-stressed individuals cooling these areas results in improvements to physiological indexes and individual perception of thermal strain [64]. In the horse it is suggested that such thermally sensitive areas are represented by the poll area and ventral parts of the neck.

The author has used cooling collars for several years and anecdotally they appear to be a promising therapeutic strategy for EHI in TB racehorses. The collar appears to alleviate signs of distress after racing and further research into cooling collars in racehorses is warranted. It must be emphasized, however, that the collar is recommended only for use as an adjunct to whole-body cooling and does not replace it.

#### 4.4.2. Whole Body Cooling Strategies

There are a multitude of articles concerning cooling strategies in horses [65,66,67]. Rather than providing a critical review of those articles, this overview aims to provide an understanding of the scientific principles governing heat transfer and their practical application to cooling procedures at the racetrack. The recommendations herein are the result of experience with actual cases of EHI in racetrack settings. 


**
*Critical Point 1: The Determinants of Heat Transfer—The Importance of the Cooling Power of the Environment and the Use of Skin Surface Temperatures (SSTs) to Prioritise Horses for Cooling*
**


Heat is transferred by diffusion through the blood vascular system and by conduction from deep tissues to the skin surface along temperature gradients from high to low. Dissipation of that heat will then depend upon the temperature difference between the skin surface and the surrounding air. 

When horses race in cool environments, say, 10.0 to 20.0 °C, their skin temperature at 30.0 to 33.0 °C [68] is greater than air temperature and heat will be readily dissipated to the environment. This is called dry heat loss. If, however, the air temperature is higher than skin temperature, say, >36.0 °C, dry heat loss is not possible and instead heat may be gained from the environment. It is at this point that heat loss becomes dependent upon the production and evaporation of sweat. Air movement is extremely important, providing additional convective and evaporative cooling which can temper SST and mitigate heat dissipation. Alternatively, wind-still conditions will impair the cooling process and predispose individuals to EHI.


**
*Practical Points*
**


(1)Track veterinarians (TVs) must understand the prevailing environmental conditions on any race day. These dictate the efficiency of thermoregulation and are the cornerstone of risk assessment. A recent article by the authors outlines the impact of environmental conditions, specifically on TB racehorses [69].(2)Measurement of skin surface temperature (SST) gives an indication of the potential efficiency of heat transfer from a heated core to the skin surface and thence to the environment. It allows early detection of horses at risk of EHI, enabling prioritisation for aggressive cooling. SST can be assessed objectively using the hand-held infrared thermometer **^d^** (IRT) [68]. An SST > 39.0 °C indicates a ‘hot skin’ and enables targeting of that individual for aggressive cooling. During the IRT study, SST values of 40° to 42.0 °C were not uncommon. It needs to be made clear, however, that IRT values are not a substitute for rectal temperature and are a distinctively different data set [69]. Alternatively, an experienced racetrack worker can be educated to simply feel the skin surface with the back of the hand and assess the need for a cooling intervention (see Figure 14 and Figure 15).

**d** Digitech QM7221 InfraRed Thermometer, Instrument Choice, Cavan Road, Dry Creek, South Australia 5094.


**
*Critical Point 2: The ‘Golden Time Frame’ between first detection and treatment identifies level of risk of heat injury, because the ‘Degree Minutes’ which an individual endures will dictate the associated thermal injury*
**


The concept of the ‘golden time frame’ was introduced from experimental studies in rats [70], where it was first reported that HS outcomes were reliably predicted by the ‘degree-minutes’ above a certain temperature. Researchers and clinicians working in the human athletics and military fields agreed that the time between the onset of EHI and commencement of treatment was the most important factor determining successful, no-complication outcomes. For humans, the vague neuropsychiatric symptoms are easily missed, often resulting in an unknown start to the ‘golden time frame’. Similarly for TB racehorses, the Level 1 clinical signs [32] are also easily misdiagnosed or disregarded.

Clinicians managing cases of EHI in human athletes aim to decrease T*c* below 40.5 °C within 30 min from first presentation. This benchmark is referred to as the ‘critical thermal maximum’ for heat injury, and for humans has historically been placed in the vicinity of 40.5 °C [71] (see Figure 16). For horses and dogs, it is suggested that the benchmark is considerably higher. 


**
*Practical Points*
**


(1)The treatment directive for EHI is early detection, rapid assessment, and aggressive cooling.(2)A short first-recognition-to-treatment period is the most important determinant of a successful outcome.(3)Drugs and cooling devices must be available at short notice, and on hot/humid days an experienced worker should inspect every horse after racing for levels of distress and early signs of EHI.(4)TVs must be aware that once a horse is recognised as being heat affected, all systems need to be mobilized to achieve the fastest result.


**
*Critical Point 3: Horses can be aggressively water cooled using various techniques, but whatever the technique, the fastest cooling rates will be achieved using ice-cold water*
**


In accordance with the principles of heat exchange, the rate of transfer of heat energy from the skin to the cooling medium (water) will be proportional to the temperature difference between the two [13], thus the importance of ice-cold water. Cooling the skin draws heat from the underlying tissues, extending the temperature gradient inwards. 

Special circumstances exist with heat-affected horses. The combination of strenuous exercise and adverse weather conditions results in substantial increases to skin blood flow and reductions to cardiac output, creating a redistributive hypovolemia [30,31]. Because the rate of convective heat transfer is determined by the product of blood flow and the temperature gradient between core and skin, the latter will largely dictate the rate of heat transfer in such conditions. Cooling the skin redistributes blood back from the periphery to the central circulation and decreases thermal strain.

***The various cooling techniques using water will differ in efficiency, but their ultimate performance depends largely on the temperature gradient between the water and hot core*** [72]

**(1)** Complete water immersion (CWI) represents conductive cooling. Human athletes with EHI are placed in large tubs of ice-cold water. This technique has significant practical limitations with large animals like horses but works extremely well in cooling greyhounds after racing (see Figure 17).**(2)** Spray cooling heat transfer is the closest to the conduction cooling technique and works well for horses. The intention is to spray an even layer of water over as much of the skin surface area as possible (see Figure 18). A methodical approach is utilized, whereby water is applied first to the head and neck, then the chest and forelimbs, followed by the hindquarters and between the back legs. The superficial veins, being the major heat transporters from core to skin, are directly targeted. Once one side has been traversed, the operator moves to the other side, or alternatively, two operators can spray both sides simultaneously. The aim is to replenish the cool water in a consistent cycle. The use of specially designed hose nozzles**^e^** can be particularly useful. They are easily adaptable to all hoses, ensure even coverage of the skin surface due to uniform droplet size, can be selected for spray pattern (full cone or square), allow adjustment of the spray characteristics from medium to coarse atomisation, and are designed to work with variable flow rates. This versatility increases the efficiency of the spray cooling technique.

**e** Spray Nozzle Engineering—Total Spraying Solutions, 1-8/27 A Shearson Crescent, Mentone, Victoria 3194 Australia. sraynozzle.com.au

**(3)** **‘Dousing’.** With this technique water is applied indiscriminately over the horse’s entire body, usually using fixed, open hoses within an enclosed space. Some racing jurisdictions have ‘dousing’ stalls where horses exhibiting signs of EHI can be placed and showered continuously until their clinical manifestations resolve (see Figure 19). The limitation to this technique is that large volumes of water are used and targeting of the major vessels is not possible.**(4)** The bucketing of water over the body of horses cannot be recommended because the weight and concentrated volume causes most of the water to fall to the ground, reducing the efficiency of cooling (see Figure 20).


**
*Practical Points*
**


**(1)** To facilitate the practical use of ice-cold water, a barrel or tank that can be filled with ice is a prerequisite. A mobile spray unit used by the first author in Australia consisted of a trolley-mounted 180-litre plastic tank with a large screw cap for ease of ice loading. A 12V battery-powered pump and retractable hose connected to a trigger-operated spray nozzle gave the operator complete control over the flow rate and distribution of the ice water. The author (MB) believes these units transformed the treatment of EHI at the racetrack. If a difficult day was expected, several units would be prepared and stationed at strategic positions. Most horses succumb to an EHI event immediately after racing, so the dismounting yard and the tie-up stalls are logical locations. It is essential that the units are mobile, so that wherever a horse experiences an EHI event, the device can be easily moved. The designs of cooling devices are many and varied (see Figure 21, Figure 22 and Figure 23). The major practical consideration in their use is the availability and continual supply of ice, and this needs to be the responsibility of race-day organizers as an absolute welfare priority, especially in the hot summer months.**(2)** ***When treating horses with EHI, the cooling needs to be continuous and*** uninterrupted until all signs of CNS dysfunction disappear and the animal becomes aware of its surroundings. Interrupting cooling to walk ataxic horses is not recommended.**(3)** ***What is the optimum cooling endpoint?*** It is inadvisable to target a specific temperature to cease cooling but rather to use resolution of CNS dysfunction and normalization of behaviour. The IRT can be used to great advantage and when SSTs decrease to around 30.0 °C and cutaneous skin vessels are beginning to disappear (see Figure 24), cooling can be stopped. These horses can then be walked but should be kept under scrutiny for a further 30 min. Applying the cooling collar or placing them in a shaded stall with access to fanning are both beneficial initiatives at this stage. It is most important that these horses are not left unattended in the tie-up stalls.**(4)** ***What is rebound hyperthermia***? Because of the high mass to surface area ratio of the horse, it is possible to cool the shell but not the core. Horses that are very hot in the post-race period may need to be cooled for up to 15 to 20 min. They can then be allowed to walk but still need to be under veterinary supervision, because they may experience redistribution of stored heat from the core. The cooling collar has been found most useful in these cases, but continued monitoring will still be required.**(5)** ***Is scraping necessary?*** There is substantial controversy concerning the scraping of excess sweat or water off the coat of horses in the post-exercise period, because it is thought that if this layer of fluid is not removed it will prevent the horses from cooling adequately. Veterinarians must consider the evaporative capacity of the environment for guidance. If it is hot and there is air flow, there will be no need to scrape, because the surface water or sweat will disappear due to the efficient evaporative capacity of the environment. If, however, it is humid with no air flow, the skin surface will remain wet and under such circumstances it is best to scrape excess water off the animal’s skin (see Figure 25). In recent studies it has been shown that not scraping during the cooling process did not cause core temperature to rise [67].**(6)** ***The provision of shaded areas***. This is an important welfare initiative, especially if racing takes place during daylight hours, because it diminishes the direct effects of radiant heat on the animal’s body. The Hong Kong Jockey Club has an excellent arrangement for horses to be cooled immediately they return to the dismounting yard (see Figure 26). Shade cloth covers almost the entire area, artificial grass does not reflect heat as bitumen surfaces do, long hoses are strategically placed to provide cool water, and dry fans are located at many points, producing an air flow of up to 3.5 metres per second. Horses are usually at higher risk of EHI in the immediate post-race period, and an area such as this allows veterinarians to scrutinise the whole field while horse handlers initially cool their horses. This enables detection of abnormal behaviours and targeting of an individual for supervised veterinary care if required.


**
*Critical Point 4: The Addition of Evaporative Cooling to the Overall Technique*
**


**Dry fans** are often used in combination with the spray cooling technique. The basic scientific principle is that increasing the air movement around the body increases the convective/evaporative heat loss to the environment, especially when the air temperature is lower than skin temperature. However, once the temperature approximates or is greater than skin temperature, dry fans may become counterproductive, adding to heat gain by forced convection [73,74].

**Misting fans.** There are generally two types. They can either provide a fine spray of moisture through an articulated array of fine nozzles, as shown in Figure 27, or be combined with air-driven systems at high pressure, which generate smaller droplets of water and are more effective in wetting surfaces. By combining various rates of air flow, misting fans assist the evaporation of heated moisture from the skin surface [75] but they also add water to the air, reducing its evaporative capacity. In a study of heat mitigation methods in army trainees [76], one researcher found that in hot and humid environments core body temperature continued to rise while the misting fans were operating, and another reported that the humidity level close to misting fans was increased by up to 20% [77].

## 5. Discussion

This review has been written specifically for veterinarians working with TB racehorses. Although the incidence rate of EHI/HS in racehorses globally is unknown, a study by the first author [36] gave an overall incidence of 9.5% (457 cases out of 4809 starters). Other studies provided much lower incidence rates: 0.13% in South Africa [2] and (0.04%) from Japan [4]. The Japanese authors, however, declared that the incidence level had risen to over 0.7% in the past four years and attributed that rise to an increased awareness of the condition. It is suggested that the reason for such differences lies firstly in the recognition and recording of the earliest clinical signs, referred to as Level 1 by the authors. In horses, as in human athletes, these earliest signs of CNS dysfunction are typically quite vague and may go unnoticed or be attributed to other post-racing disorders. They may also resolve when routinely treated, without ever progressing to higher levels of the disorder. Consequently, incidence rates may reflect the diligence and/or experience of different veterinarians in early detection reporting. It is possible that both the South African and Japanese studies have based their diagnoses of EHI on the more obvious signs of CNS dysfunction, observed at Level 2 and above. Secondly, it is important to appreciate that EHI incidence rates will depend on specific climatic conditions or geographical contexts, and incidence figures are not transferable to other countries or even between one racetrack and another in the same vicinity. 

In a recent analysis of risk factors associated with cases of sudden death in TB racehorses in North America and Canada over a period of 12 years [78], it was reported that ‘post-exertional distress’ (PED)/(HS), which is an older alternative terminology for EHI, was identified in 21 or 3.9% of cases. This figure was equivalent to risk factors associated with sudden death from exercise-induced pulmonary haemorrhage (also 21 and 3.9% of cases) and was greater than cases attributed to lethal arrythmias (12 and 2.2% of cases). Considering that these figures represent deaths only, what might the EHI incidence rate have been for horses which were clinically affected but survived? These statistics suggest that cases of EHI cannot be dismissed as irrelevant and that incidence figures are probably quite high, but largely misdiagnosed and under-reported. 

Research into EHI is difficult because it is unethical to induce the condition in the laboratory in humans or animals. This means that randomised controlled trials for treatment strategies are not available, and we must rely on anecdotal reports from studies such as that cited above [36], performed at the racetrack with actual EHI cases. 

The most important welfare initiative concerning EHI in TB racehorses is education at all levels. Race-day veterinarians, horse handlers, horse trainers, swab officials, and race-day operating personnel should all be aware of the condition and its treatment requirements. The creation of a cooling infrastructure, incorporating specialised stalls, shaded areas, mobile cooling devices, installation of fans, availability of hoses and suitable nozzles, and extra staff on ‘hot’ days, is a welfare priority and should be driven by veterinarians with the full co-operation of racing organizations. If climate change continues to develop, there will be a greater number of hot days when horses must compete in adverse weather conditions and EHI/HS may well become racing’s most important welfare issue. 

## 6. Conclusions

This article has concentrated on diagnosis of EHI and its treatment in TB racehorses. Early detection is the cornerstone of effective treatment and the authors have outlined in some detail precise case definitions describing the various levels of CNS dysfunction. A recent consensus statement from experts in the human field has revealed a shift away from reliance on a temperature threshold for EHI diagnosis, and levels of CNS dysfunction are now considered more likely to define the severity of the condition with greater sensitivity and specificity. 

The unique thermoregulatory capacity of the racehorse is emphasised because confusion can occur in the immediate post-race period, between what is normal after an arduous run under adverse conditions and what is emergent EHI. Particularly relevant is post-race redistributive hypovolemia, a low cardiac output state where blood volume is sequestered in the compliant peripheral circulation. Reduced blood flow means that heat transfer from a hot core to the skin surface becomes more dependent upon the existence of a wide temperature gradient, validating the use of ice-cold water as the superior cooling medium. Similarly, the concept of the critical thermal maximum informs the practitioner that the time taken to cool directly relates to the degree of tissue injury incurred, and again this supports the use of ice-cold water as the superior cooling modality. The technique employed is of less importance than the use of this low-temperature coolant.

Understanding the pathophysiology of EHI is essential, and the perspective now includes a gastrointestinal endotoxemic pathway, where the immune status of the host may predispose to the EHI condition. A totally heat-centred approach does not explain inconsistencies like the occurrence of EHI on cooler days, or the fact that heat stroke pathology is more like that seen in bacterial sepsis rather than the singular effects of heat toxicity. It is apparent that EHI/HS needs to be regarded as a more complex medical condition, where both thermoregulatory and inflammatory pathways may be operating in concert. Clarification of the existence of the two pathophysiological pathways also provides a scientific basis for the recommended pharmacological treatment options.

Finally, from the authors’ viewpoint, the game changers in the treatment of EHI are (1) efficient early detection, (2) administration of detomidine and flunixin in quick succession if CNS signs are apparent, and (3) the use of a mobile cooling device with ice-cold water as the cooling medium and finally (4) the creation of a cooling infrastructure at racetracks as an absolute welfare priority.

## Figures and Tables

**Figure 1 animals-13-00610-f001:**
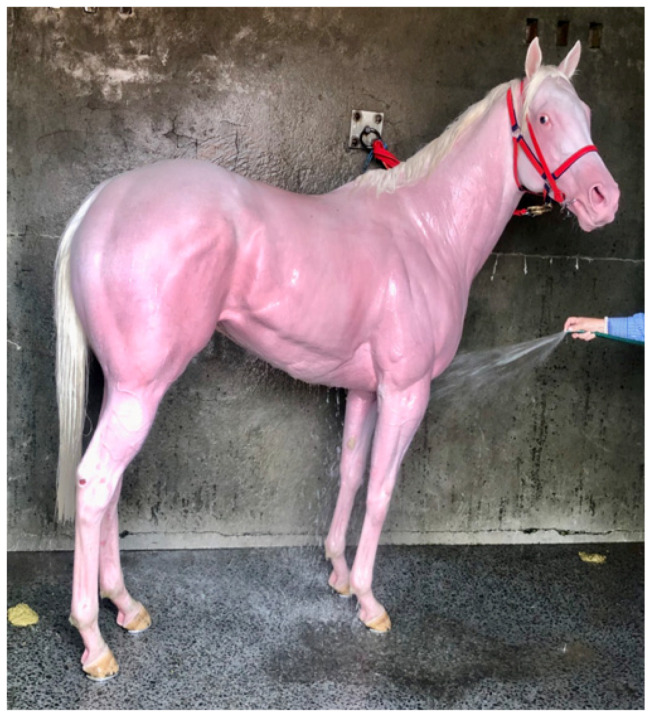
Albino horse in the immediate post-race period on a hot day. The pink colour indicates that almost the entire skin surface has increased blood flow for heat dissipation. Source: J Equine Vet Ed 2021.

**Figure 2 animals-13-00610-f002:**
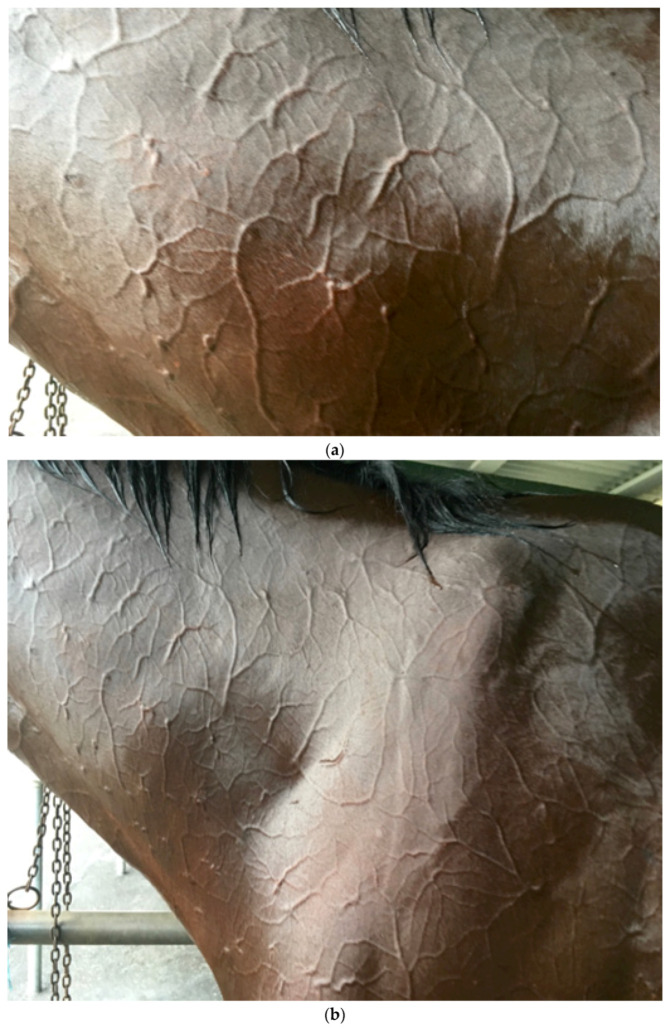
(**a**,**b**) shows the delicate pattern of vessels covering all the visible skin surface. Specialised anatomical structures called arteriovenous anastomoses (AVAs) connect arteries directly to veins, facilitating increases in peripheral blood flow without traversing capillary beds, permitting rapid and efficient heat loss [13]. Source—J Equine Vet Ed, 2021.

**Figure 3 animals-13-00610-f003:**
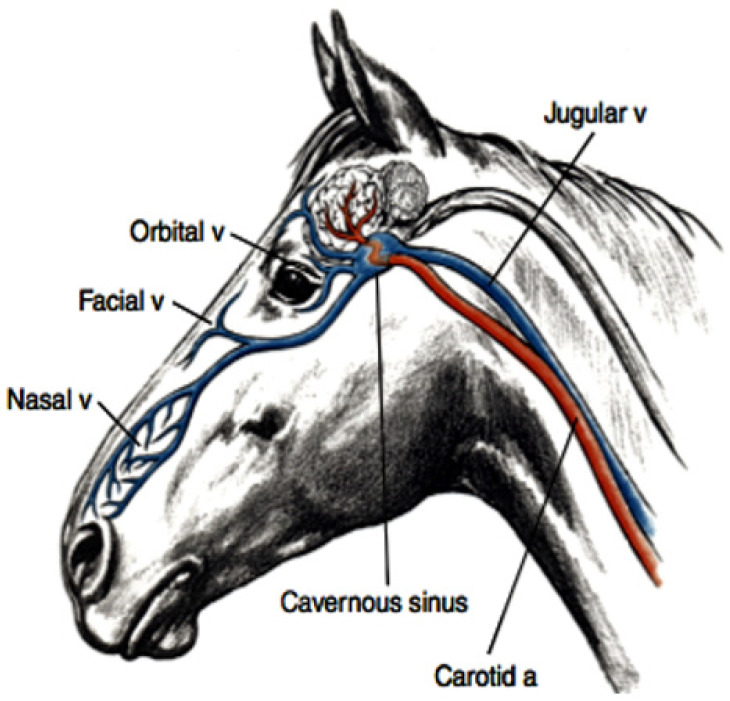
Diagram showing the anatomical arrangement for evaporative respiratory heat loss. The complex system of turbinate bones in the upper respiratory tract is covered by a mucous membrane with a rich blood supply and numerous AVAs, so that when blood enters the nose on inhalation it is cooled and, when necessary, gains direct access to the venous side of the circulation via the AVAs. Venous transporters such as the angularis occuli referred to as orbital vein in this diagram, drain the face and scalp and flow into the cavernous sinus at the base of the brain. The facial vein transports cooled blood from the upper respiratory tract directly into the jugular vein, thence to the right heart and central circulation, effectively contributing to whole body cooling. Diagram with permission from Dr. Finola McConaghy, PHD thesis 1995. (Blue represents venous blood; red represents arterial blood).

**Figure 4 animals-13-00610-f004:**
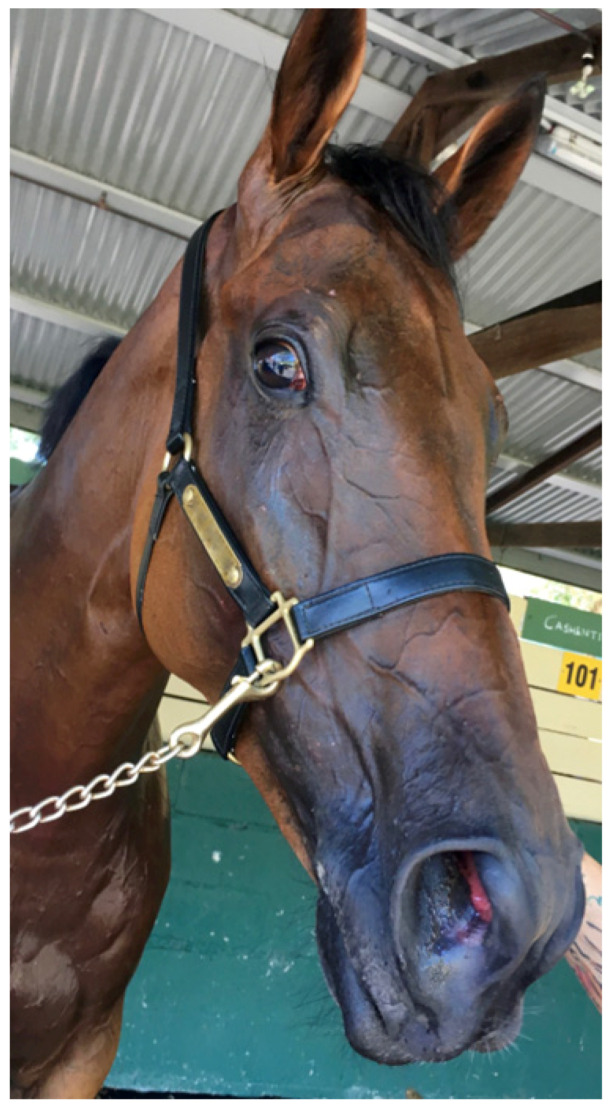
This horse finished racing 10 min previously and exhibits a ‘panting’ type of respiration. Note the widely dilated nostril and the dilated venous transporters which drain the scalp and face, typical of horses that are working hard in the ambient conditions to thermoregulate. Such horses should be monitored carefully and cooled again until they stop panting. Source—J Equine Vet Ed 2021.

**Figure 5 animals-13-00610-f005:**
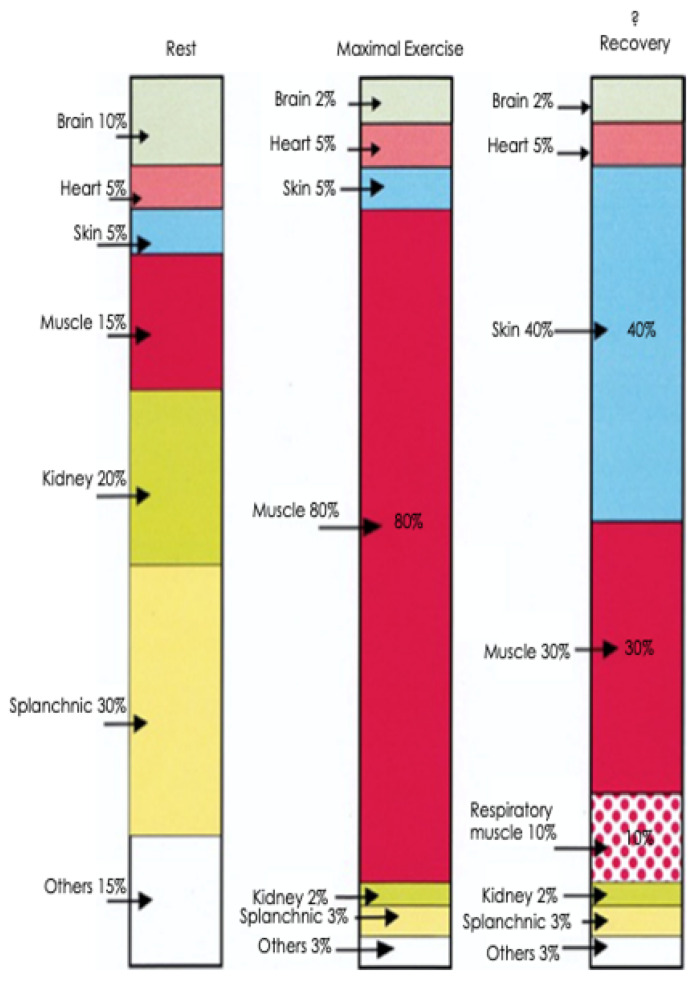
Distribution of cardiac output at rest, during maximum exercise and into the recovery period. During strenuous exercise it is highly likely that muscle blood flow may reach 80% of cardiac output, whilst skin blood flow will diminish to only 5%. Also note the substantial reductions in splanchnic and renal blood flow to 3% and 2% respectively. In the recovery stage there are still requirements for muscle blood flow, but the skin assumes major importance as the major recipient of cardiac output because of its major role in heat dissipation by sweating. Note the suggested increase in respiratory blood flow during the post-race period. This diagram was adapted from Poole and Erickson [23] and used with permission. The bar graph representing the recovery period was suggested by the senior author in consultation with D. C. Poole [31].

**Figure 6 animals-13-00610-f006:**
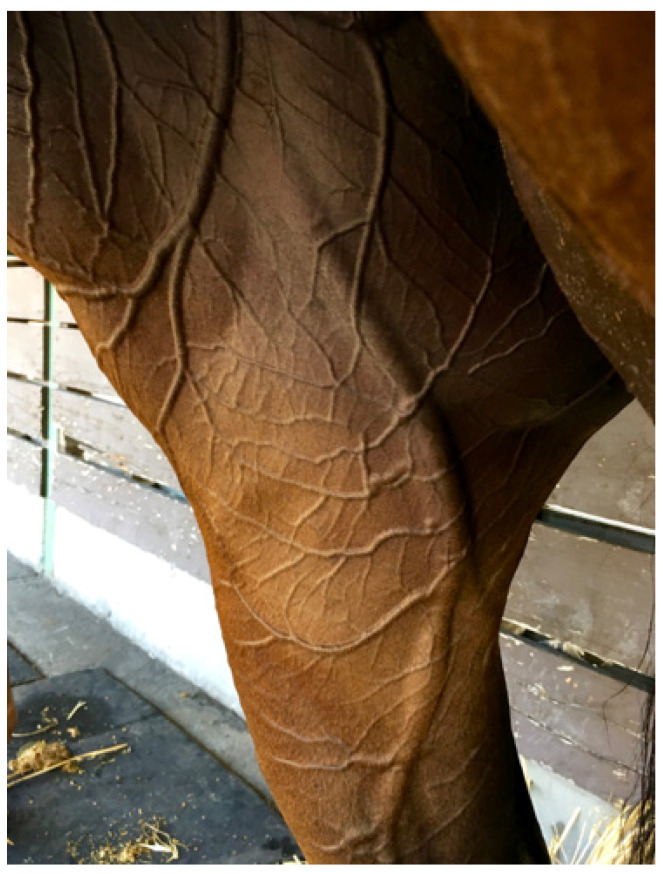
The medial aspect of the hindleg of a horse some 15 min after racing. Note the intricate pattern of small vessels and the large dilated saphenous vein minutes after racing [31]. For the track veterinarian, a knowledge of the physiology of the recovery period is of crucial importance because this is the time when horses are at greatest risk of EHI, and it is sometimes difficult to discern between physiological changes resulting from a very arduous run and those due to emergent EHI. Source—J Equine Vet Ed, 2021.

**Figure 8 animals-13-00610-f008:**
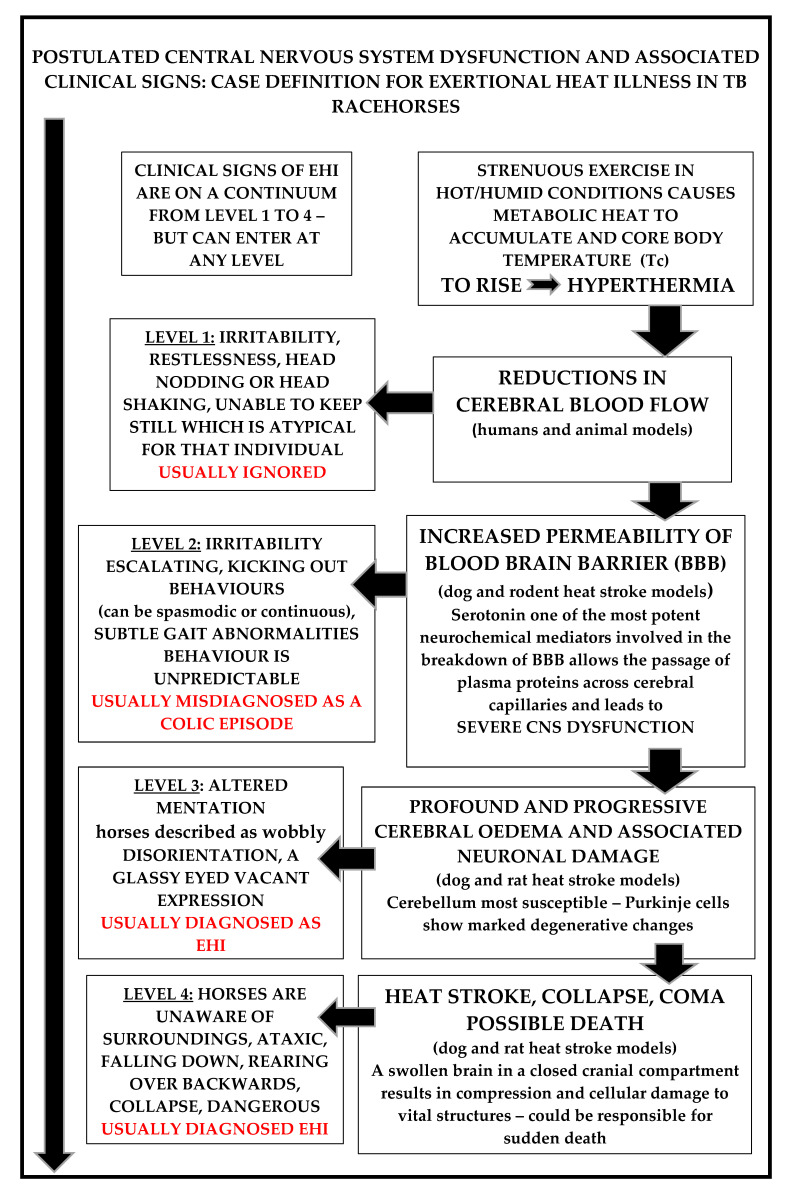
Case definition of EHI in TB racehorses and the postulated associated CNS pathology. Note that Levels 1 to 4 are on a continuum, but horses may enter at any level. Level 1 is the most important to recognise and is the most often misdiagnosed. Early recognition of EHI is essential because early treatment interventions enable the best outcomes [32,34,35,36,46,47,48].

**Figure 9 animals-13-00610-f009:**
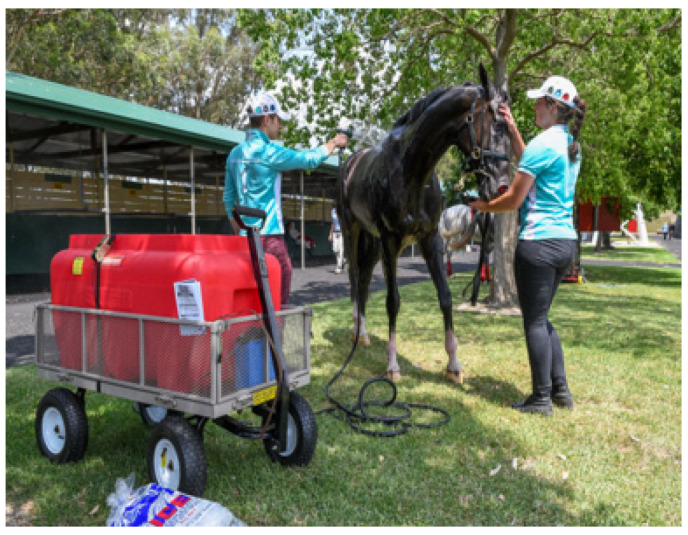
Level 1 EHI. It is difficult to show an irritable, restless horse in a still photograph. This horse has just raced and has a level of distress, with elevated heart and respiratory rates. The horse handlers have identified its requirement for a cooling intervention and have accessed the mobile cooling device, which contains ice-cold water. Veterinarians need to be always on hand to provide advice and monitor horses during the cooling process. Source—J Equine Vet Ed, 2021.

**Figure 10 animals-13-00610-f010:**
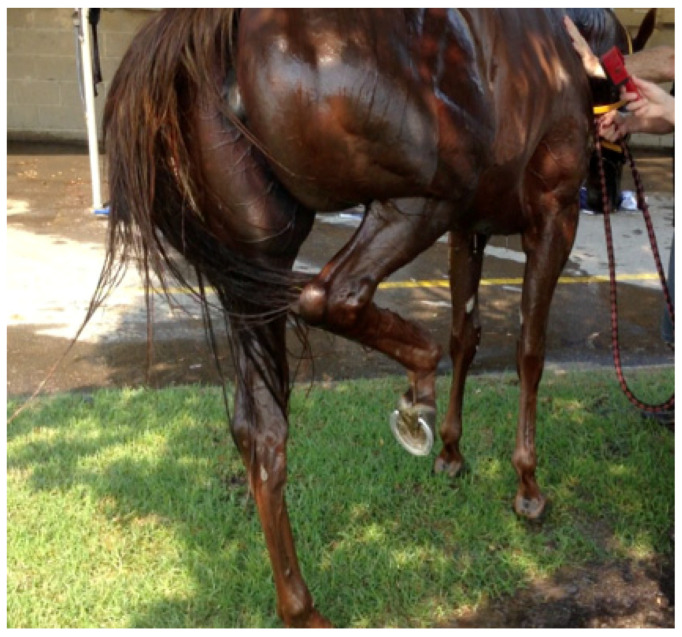
Level 2 EHI: This horse is displaying kicking-out behaviour. It may be random and only spasmodic, or continuous and quite violent, and can easily be misdiagnosed as a colic episode. Source—J Equine Vet Ed, 2021.

**Figure 11 animals-13-00610-f011:**
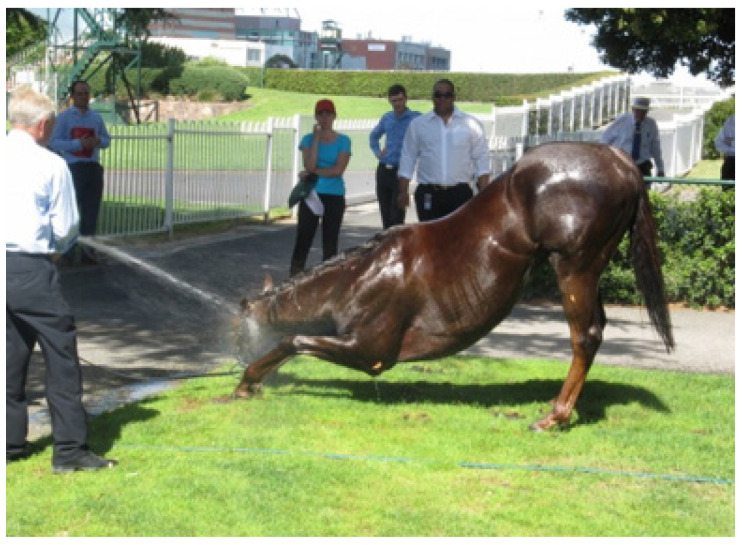
Level 3 EHI: Any bizarre CNS dysfunction will fit into this category. Altered mentation is characteristic and horses are described as having a ‘spaced-out’ or glassy-eyed’, ‘vacant expression’. They can also be dangerous to themselves and to their handlers. Source—J Equine Vet Ed, 2021.

**Figure 12 animals-13-00610-f012:**
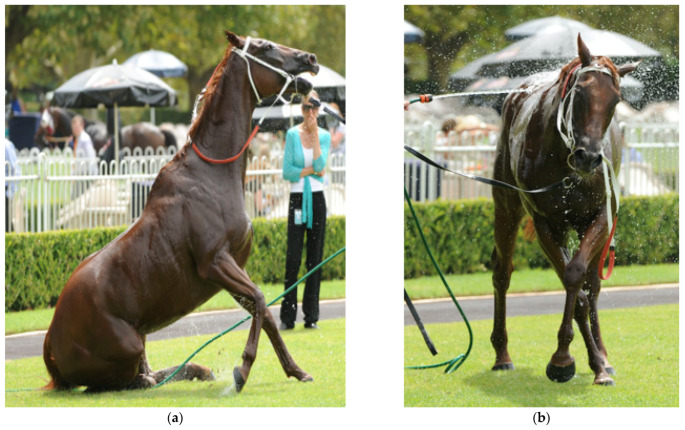
(**a**,**b**) show a horse that has just returned to the enclosure after racing. It has immediately displayed significant levels of ataxia and has altered mentation. The horse pictured in 12 (**c**) has returned to the enclosure and collapsed. In hot/humid weather EHI is an important provisional diagnosis and horses need to be cooled in the first instance before EHI can be discounted. Figure 12: (**a**,**b**) source—J Equine Vet Ed, 2021.

**Figure 13 animals-13-00610-f013:**
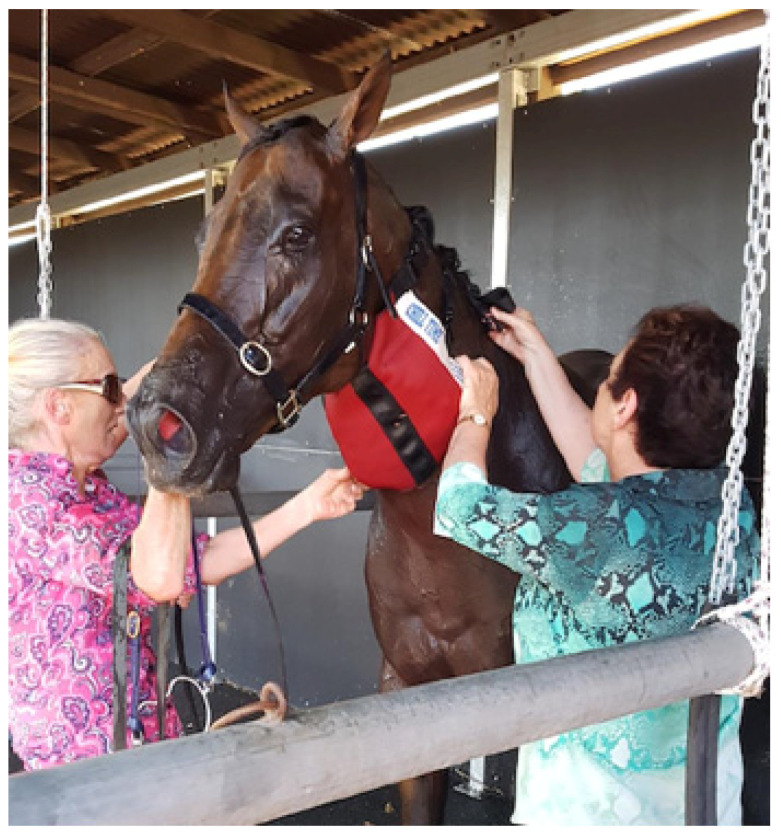
The author (MB, right) applying a cooling collar to a horse in the immediate post-race period. The horse showed signs of distress, irritability, restlessness, and was occasionally kicking out. It has been cooled with iced water and the ice collar used as an adjunct to whole body cooling. Note the dull-eyed look, slack ear position, and the dilated nostrils. Source—J Equine Vet Ed, 2021.

**Figure 14 animals-13-00610-f014:**
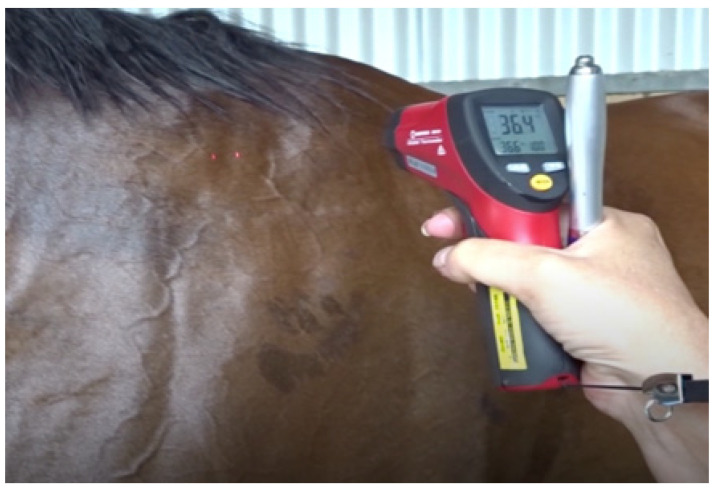
The infrared thermometer enables an objective assessment of skin surface temperature. Source—J Equine Vet Ed, 2021.

**Figure 15 animals-13-00610-f015:**
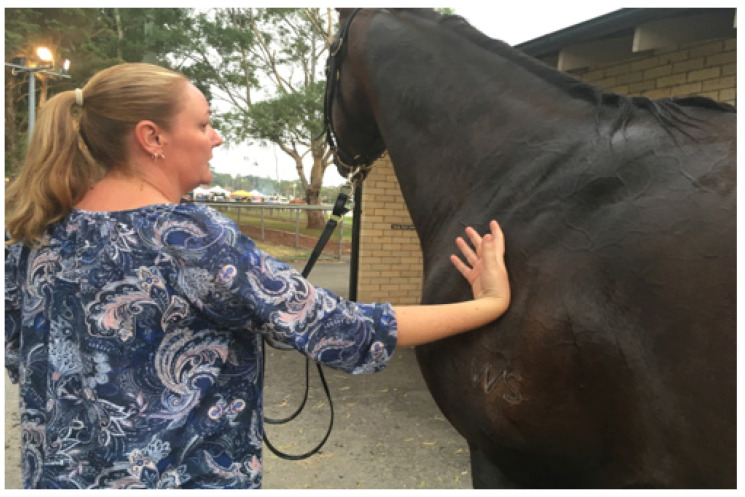
An experienced veterinary worker can subjectively assess skin surface temperature. Source—J Equine Vet Ed, 2021.

**Figure 16 animals-13-00610-f016:**
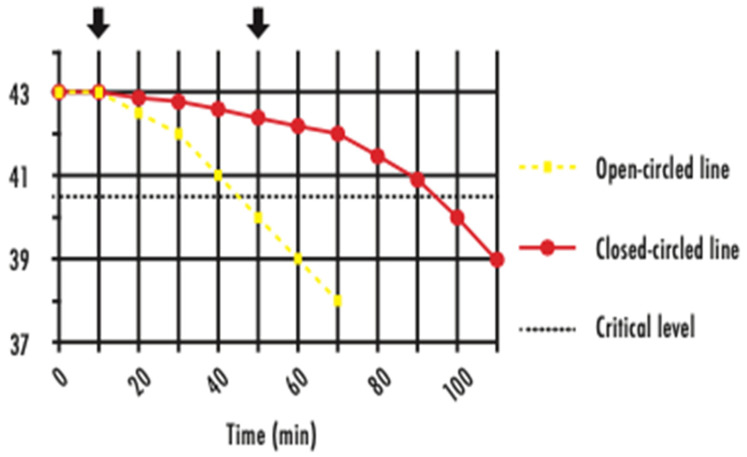
Graph showing the concept of the ‘critical thermal maximum’ for thermal injury in human subjects. The aim of cooling is to decrease the T*c* below the critical level (dotted line), considered to be 40.5 °C, in the shortest possible time. An early intervention (yellow) achieves a rapid decrease in T*c*, but a later intervention (red) allows T*c* to remain elevated for an extended period [71]. Source—J Equine Vet Ed, 2021.

**Figure 17 animals-13-00610-f017:**
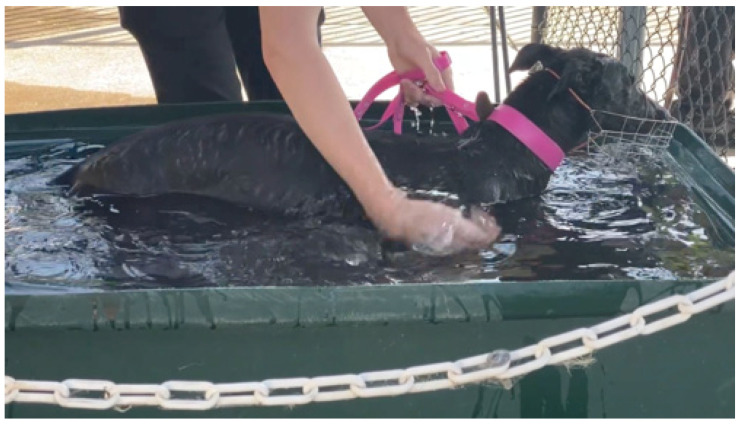
A greyhound in an immersion tub of cold water immediately after racing.

**Figure 18 animals-13-00610-f018:**
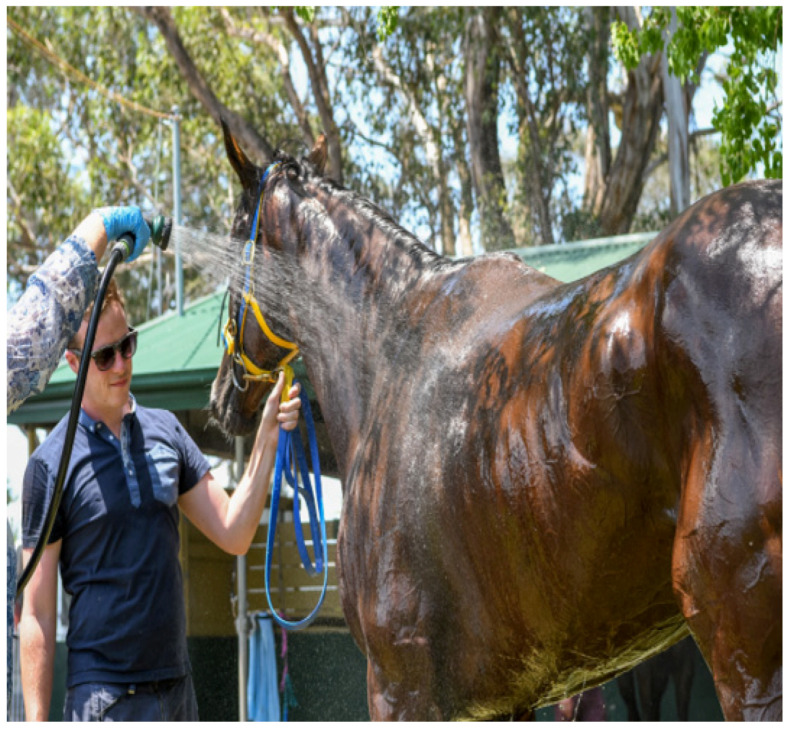
A horse being spray-hosed with ice-cold water. The nozzle type is important because uniform droplet size will guarantee a more even coverage of water.

**Figure 19 animals-13-00610-f019:**
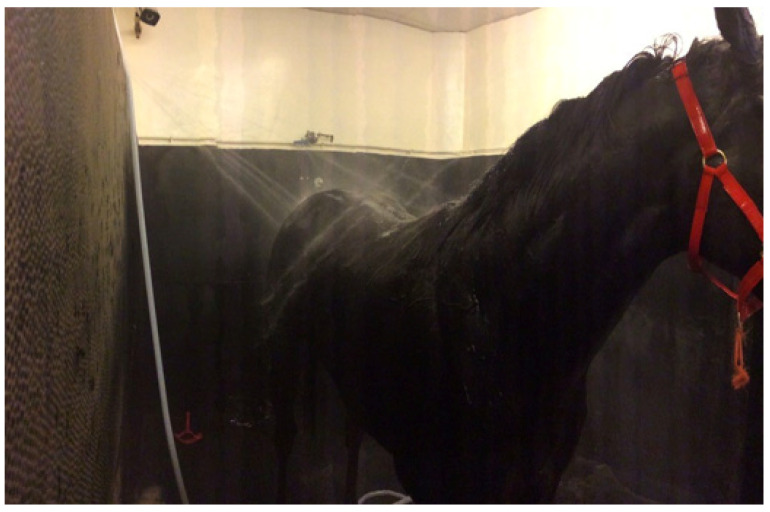
The dousing stall at the Hong Kong Jockey Club Sha Tin Racecourse.

**Figure 20 animals-13-00610-f020:**
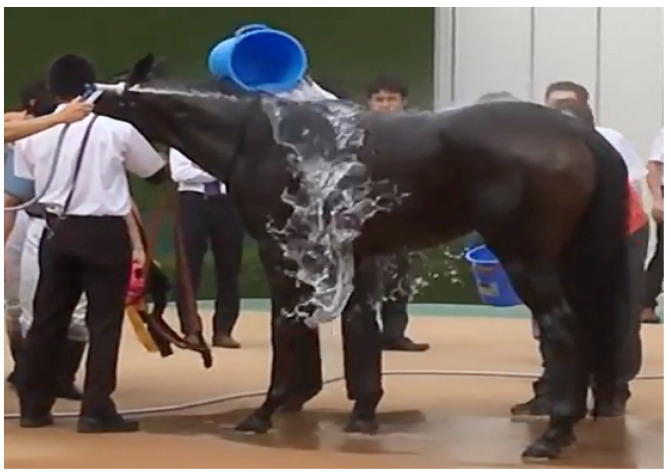
Bucketing water over horses should only ever be a last resort. As can be seen in this photograph, most of the water does not contact the skin surface for any length of time.

**Figure 21 animals-13-00610-f021:**
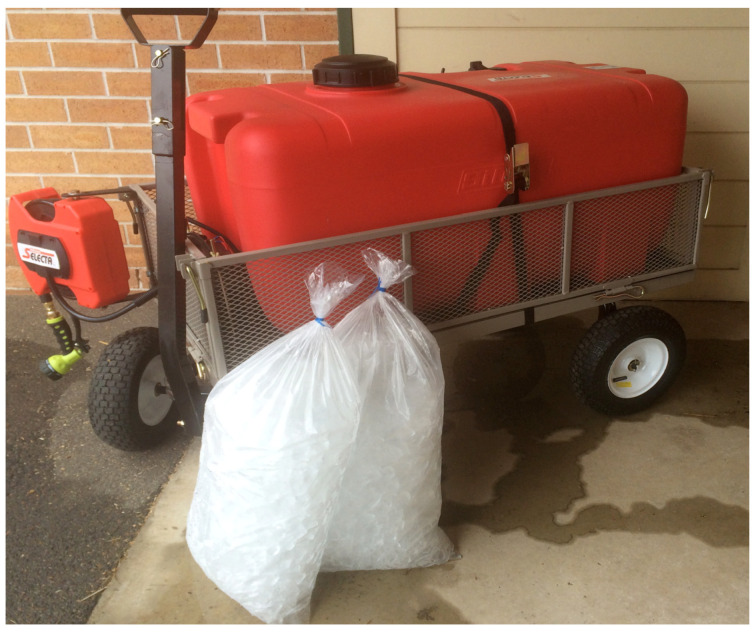
A compact, mobile cooling device. The design could be improved by the inclusion of an insulating blanket to decrease ice loss in the hot summer months and a wider filler neck to admit solid ice more easily.

**Figure 22 animals-13-00610-f022:**
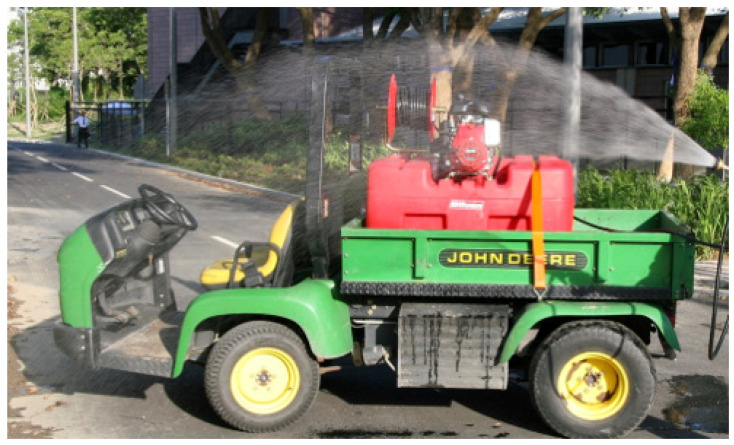
A cold-water spray unit in use at the Hong Kong Jockey Club.

**Figure 23 animals-13-00610-f023:**
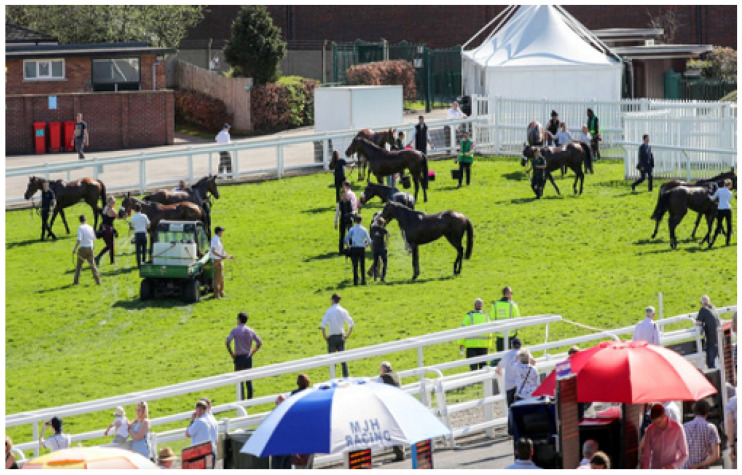
A mobile cold water spray unit in use after a race in the UK.

**Figure 24 animals-13-00610-f024:**
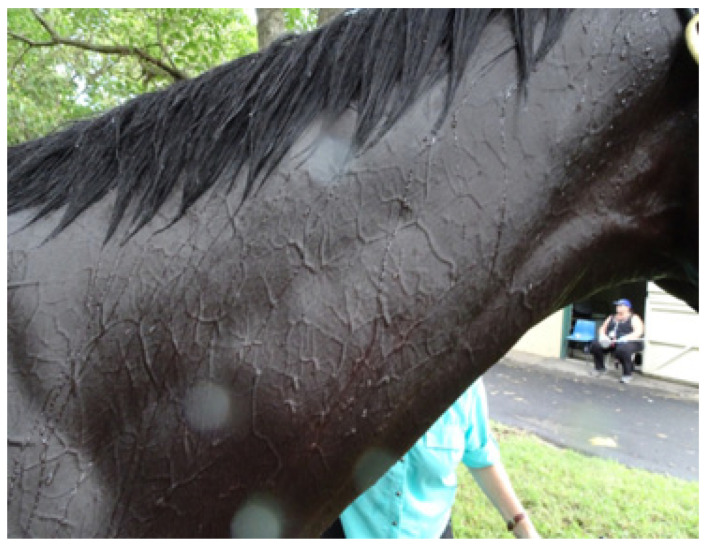
Horses should be cooled until their CNS dysfunction normalises, their skin vessels are just beginning to disappear, and their SST is decreasing.

**Figure 25 animals-13-00610-f025:**
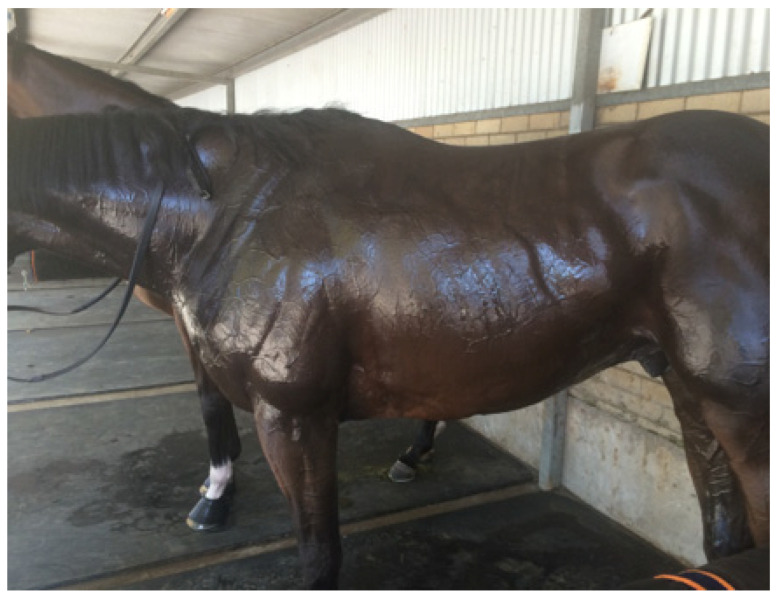
This horse is covered in sweat which is not evaporating because of the environmental conditions on the day and would benefit from being scraped. Source—J Equine Vet Ed, 2021.

**Figure 26 animals-13-00610-f026:**
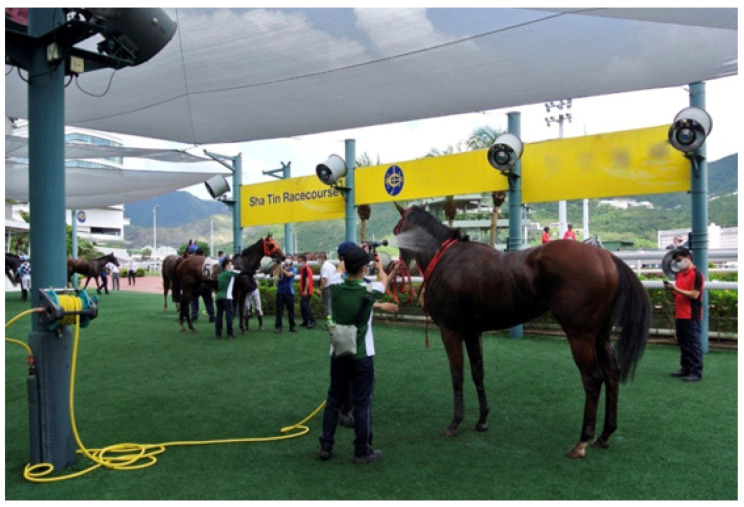
Dismounting enclosure at the Hong Kong Jockey Club, Sha Tin Racecourse. Courtesy of Dr. Peter Curl. The creation of a cooling infrastructure at racetracks such as shown in this photograph is ideal and an important welfare priority. Note the extensive shade cloth, green floor surface, mounted dry fans and multiple long hoses providing cool water.

**Figure 27 animals-13-00610-f027:**
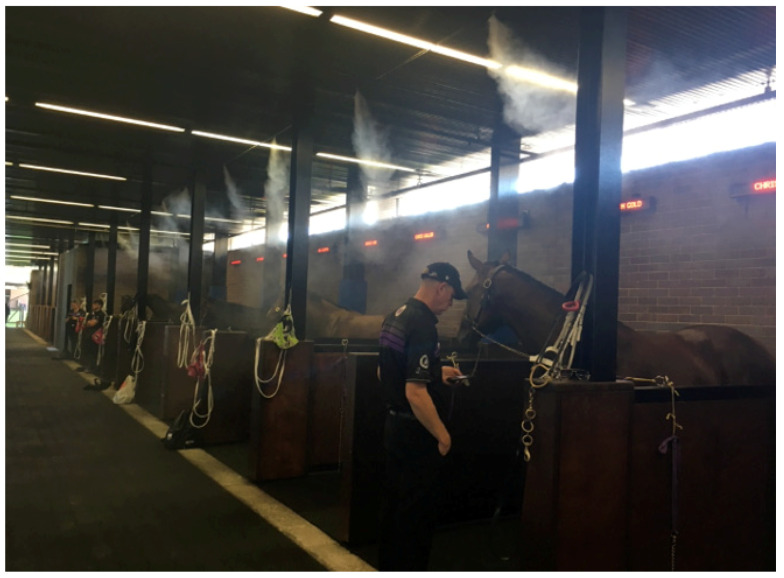
Misting sprays in the tie-up stalls at a racecourse in Australia. The devices can theoretically increase the humidity by up to 20% in the immediate vicinity, negatively affecting thermoregulatory capacity.

## Data Availability

See: Brownlow, M.A., Brotherhood, J.R. An investigation into environmental variables influencing post-race exertional heat illness in thoroughbred racehorses in temperate eastern Australia. *Aust. Vet. J.* **2021**, *99*, 433–481.

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
