# Peer review of "An Overview of Exertional Heat Illness in Thoroughbred Racehorses: Pathophysiology, Diagnosis, and Treatment Rationale"

_animals, 2023, doi:10.3390/ani13040610_

Round 1
Reviewer 1 Report
the authors provided an exhaustive overview of EHI/HS in race horses, focusing on clinical signs detection and more effective therapies. Being the major audience the race track veterinarian, the major keypoints of the paper are pointed to the diagnosis and therapy rather than pathophysiology, and I truly appreciated this.
The authors already showed an extensive knowledge of the topic, as proved by more recently published papers.
I only have to advice some minor revision.
line 29-40: please check the character dimension
line 47: I can't find reference n.1
line 149: bold character? check if appropriate
line 209: some of the sentence is missing
line 222: the beginning of the sentence is missing
line 231: check the sentence. It seems incomplete
line 250: some of the sentence is missing
line 253: does it refers to fig. 6
line 306: some of the sentence is missing
fig 7: I don't understand the figure/scheme, please clarify
line 663 and 740: should this be expressed as foot note?
line 808: some of the sentence is missing
Author Response
- I have changed character dimension as required
- Reference 1 inserted on Line 16
- Suggestion from reviewer 2 to put must of the subheadings in bold and italics?
- There has been an apparent problem with this edit-version of the article - where there are figures and their position within the manuscript have causing some of the sentances to re-arrange themselves. This accounts for line 209, line 221, line 231, 250,306,808. I have been through the manuscript thoroughly and I beieve the problem is solved as long as the figures stay in place. It may be best to view the PDF version if there are problems.
- Line 253 does refers to critical thermal maximum for discussion see Figure 16.
- Figure 7 is deleted and a new figure 7 added at the back of the references.
- I am somewhat devastated that there were so many problem s with sentances due to changes to image dimensions.
Reviewer 2 Report
This comprehensive revision is focused on the Exertional Heat Illness in horses discussed on the background of the Exertional Heat Illness in humans. The article is well written and easy to follow therefore I believe that it will be useful for many recipients involved in the breeding, training, and treatment of horses exposed to an excessive elevation of core body temperature.
Substantively, the content of the article is very good and much needed to be published, therefore my detailed comments below are of an editorial nature.
Detailed comments:
In the abstract section, please use the same font size, remove enter mark between L 28-29, and expand HS and TB.
L 53 remove the additional space mark
L 61 replaces exertional heat illness with an abbreviation
L 61 expend TB (all abbreviations should be expended when used for the first time in the article and then they should be referred consequently)
L 73 remove the additional space mark
L 83 heat stroke has been introduced at the beginning of the Simple summary section however the expansion of the abbreviation
L 90 remove the additional space mark
L 91 expend EHS (EHI or HS?)
L 101 What does (p.7) mean in square brackets
L 103 remove the additional space mark
Please review the entire manuscript carefully to remove extra spaces - I will not mark them in the review - but this is needed until L 897
L 103-104 'The first author (MB) is of the view' Does that mean the second author disagrees with the first? Consider whether it would not be better to write 'The authors are of the view' or 'One should note that'.
L 69, 109, 205, 228, 268, 298, 363, 384, 462, 491, 525, and 562 decide to use colon or dot
L 111 change the colon to a dot
L 149-150 If 'Respiratory evaporative heat loss (REHL) from powerful thermo-effectors of the upper respiratory tract' is a subsection heading, it should be indicated. There is a general trouble with subhead marking in 1.2. subsection (L 111, 125 - bold and italics + colon; L 145, 179, 190 - no bolding or italics or colon; L 164, 173 - bold but no italics + coma nor colon)
Could you unify it throughout the manuscript - also in 3.4. subsection, please?
(L 563 - no bolding or italics or colon but dot and (a), L 599 - no bolding or italics, L 625 - no bolding or italics or colon but (a) at the beginning, ... and so on up to L 856)
L 146 There is a red coma left.
L 309 'The heat toxicity pathway' remove italics
L 409 Consider adding 'Level 1 – case ...' subhead in italics.
I appreciate all the work that goes into such an extensive and complete literature review, but my role as a reviewer is to draw attention to the conclusions. For the sake of clarity, they should be significantly shortened to one concluding paragraph. Please consider moving some of your current conclusions to the end of the discussion - perhaps in the form of a summary and formulating new, compact conclusions at the end.
I have just one additional question for the authors. What do you think? Since in racehorses, Infrared Thermography (IRT) has been evidenced to be useful in heat radiation measurement before and after race training, whether IRT can be useful in EHI monitoring? (DOI:10.3390/ani10112072)
Author Response
- Abstract section: same font size selected, remove enter mark, HS and TB expanded.
- Line 53 remove the additional space mark
- Line 61 expend the term TB
- Line 61 replaces exertional heat illness with EHI
- Line 73 remove additional space mark
- Line 83 expansion of HS abbreviation
- Line 90 remove the additional space mark
- Line 91 changed EHS
- Line 101 The reference to p.7 is as recommended on the instructions to authors where there is a "direct quote" such as that followed in lines from the Poole and Erickson reference (23).
- Line 103 remove the additional space mark
- Change from The first author (MB) to ........ see text in red
- Use of colon - line 68, 109, 205, 228, 268, 298, 363, 383, 462,Drugs that target CNS dysfnction 3.1:
- Drugs that target endotoxemia 3.2: Corticosteroids.3:3Note change to number. subset:
- Unification of subheadings throughout manuscript see changes n text to red
- Changes to sub-headings - italics and bold
- Line 156 Changes to heading Respiratory evaporative heat loss
- The heat toxicity pathway needs to stay in italics because the endotoxemic pathway is also in italics.
- Have have added Level 1 etc cases definition in italics - see red in text
- I have re-arranged the discussion and actualy taken some things out _see red deletions and reduced the conclusions also substantially I hope these amendments are satisfactory.
NOTE ALL SUB-HEADINGS FIRST NUMBER. SECOND NUMBER : IN ITALICS AND IN BOLD
In answer to your question concerning the use of IRT for EHI monitoring. We use this using a hand-held device and monitoring skin temperature in the post-race period. If the skin IRT level is > 39.0 degC we target those horses for immediate cooling. This means that out of a race, say 12 horses, we may select 2 or 3 for immediate supervised cooling and monitoring. Please see Reference: Brownlow and Smith (2020). The use of the hand-held infrared thermometer as an early detection tool for EHI in TB racehorses: a study in eastern Australia. J Equine Vet Ed 33, 296-305.
I am sorry but I do not know what the additional space mark is that you refer to. I just hope that the editing process fixes that up when publishing.
Many thanks for your time.